# Alternative lengthening of telomeres in childhood neuroblastoma from genome to proteome

Sabine A. Hartlieb [1,2,3,20], Lina Sieverling [3,4,5,20], Michal Nadler-Holly[6], Matthias Ziehm [6], Umut H. Toprak[1,2], Carl Herrmann [7], Naveed Ishaque [8,9], Konstantin Okonechnikov[1,10], Moritz Gartlgruber[1,2], Young-Gyu Park[1,2], Elisa Maria Wecht[1,2], Larissa Savelyeva[1,2], Kai-Oliver Henrich[1,2], Carolina Rosswog[11], Matthias Fischer [11,12,13], Barbara Hero[13], David T. W. Jones[1,14], Elke Pfaff[1,10,15], Olaf Witt[1,15,16], Stefan M. Pfister[1,10,15], Richard Volckmann[17], Jan Koster [17], Katharina Kiesel[18], Karsten Rippe [18], Sabine Taschner-Mandl [19], Peter Ambros[19], Benedikt Brors [4], Matthias Selbach [6,9], Lars Feuerbach [4] & Frank Westermann[1,2✉]

Telomere maintenance by telomerase activation or alternative lengthening of telomeres (ALT) is a major determinant of poor outcome in neuroblastoma. Here, we screen for ALT in primary and relapsed neuroblastomas ($n = 760$) and characterize its features using multi-omics profiling. ALT-positive tumors are molecularly distinct from other neuroblastoma subtypes and enriched in a population-based clinical sequencing study cohort for relapsed cases. They display reduced ATRX/DAXX complex abundance, due to either *ATRX* mutations (55%) or low protein expression. The heterochromatic histone mark H3K9me3 recognized by ATRX is enriched at the telomeres of ALT-positive tumors. Notably, we find a high frequency of telomeric repeat loci with a neuroblastoma ALT-specific hotspot on chr1q42.2 and loss of the adjacent chromosomal segment forming a neo-telomere. ALT-positive neuroblastomas proliferate slowly, which is reflected by a protracted clinical course of disease. Nevertheless, children with an ALT-positive neuroblastoma have dismal outcome.

[1] Hopp Children's Cancer Center (KiTZ), Heidelberg, Germany. [2] Neuroblastoma Genomics, German Cancer Research Center (DKFZ), Heidelberg, Germany. [3] Faculty of Biosciences, Heidelberg University, Heidelberg, Germany. [4] Applied Bioinformatics, German Cancer Research Center (DKFZ), German Cancer Consortium (DKTK), Heidelberg, Germany. [5] Division of Translational Medical Oncology, National Center for Tumor Diseases (NCT), Heidelberg, Germany. [6] Proteome Dynamics, Max Delbrück Center for Molecular Medicine, Berlin, Germany. [7] Health Data Science Unit, Medical Faculty Heidelberg and BioQuant, Heidelberg, Germany. [8] Digital Health Centre, Berlin Institute of Health (BIH), Berlin, Germany. [9] Charité – Universitätsmedizin Berlin, Berlin, Germany. [10] Pediatric Neurooncology, German Cancer Research Center (DKFZ), Heidelberg, Germany. [11] Department of Experimental Pediatric Oncology, University Children's Hospital of Cologne, Medical Faculty, Cologne, Germany. [12] Center for Molecular Medicine Cologne (CMMC), University of Cologne, Cologne, Germany. [13] Department of Pediatric Oncology and Hematology, University of Cologne, Cologne, Germany. [14] Pediatric Glioma Research Group, German Cancer Research Center (DKFZ), Heidelberg, Germany. [15] Department of Pediatric Hematology and Oncology, University Hospital, Heidelberg, Germany. [16] Clinical Cooperation Unit Pediatric Oncology, German Cancer Research Center (DKFZ), Heidelberg, Germany. [17] Department of Oncogenomics Amsterdam University Medical Centers (AUMC), Amsterdam, the Netherlands. [18] Chromatin Networks, German Cancer Research Center (DKFZ) and BioQuant, Heidelberg, Germany. [19] CCRI, St Anna Children's Cancer Research Institute, Vienna, Austria. [20] These authors contributed equally: Sabine A. Hartlieb, Lina Sieverling. ✉email: f.westermann@kitz-heidelberg.de

Activation of a telomere maintenance mechanism (TMM) was found to be associated with poor outcome in neuroblastoma[1], which is the most common extracranial solid tumor diagnosed in early childhood[2–5]. Telomerase activation is one way of maintaining telomere length and results from amplified *MYCN* (MYCN transcriptionally activates the *TERT* gene) or *TERT* rearrangement in neuroblastoma[1,6]. Activation of alternative lengthening of telomeres (ALT) presents an alternative route of telomere maintenance, which is seen in a substantial proportion of neuroblastoma tumors[1,6–8]. Mutations in the chromatin modifier *alpha thalassemia/mental retardation syndrome X-linked (ATRX)* are associated with elongated telomeres and ALT in neuroblastoma[7,8]. Furthermore, deletions of *ATRX* were associated with recurrent partial chromosomal losses in neuroblastomas without amplified *MYCN*[9]. ATRX and its complex partner death associated protein 6 (DAXX) incorporate histone variant H3.3 at pericentric and telomeric repeat sequences[10–12]. ALT activity is strongly associated with somatic *ATRX* mutations in different tumor entities[13–16]. Although loss of *ATRX* alone does not induce ALT, ATRX acts as an ALT suppressor and overexpression of *ATRX* in *ATRX*-mutant cells represses ALT markers[17,18]. Interestingly, knockdown of *DAXX* abolishes the ALT suppressive effect of *ATRX* overexpression in *ATRX*-mutant cells highlighting the importance of both complex members[17,18].

The clinical neuroblastoma risk stratification systems used by most national and international consortia (Society For Pediatric Oncology and Hematology-GPOH, International Society of Paediatric Oncology Europe Neuroblastoma Group–SIOPEN, Children's Oncology Group-COG), as well as the International Neuroblastoma Risk Group (INRG) staging system, categorizes neuroblastomas mainly based on INSS stage, age, and *MYCN* status into low risk, intermediate risk, and high risk[19]. Thus in clinical practice, ALT-positive high-risk tumors are treated with the same treatment protocol as *MYCN*-amplified and *TERT*-rearranged tumors presuming similar biological features as these telomerase-activated tumors[1].

Focused studies on ALT in neuroblastoma are rare and previous observations were mostly based on genetic sequencing studies and *ATRX* mutation status as ALT marker[6–8]. Here, we aim to deepen the knowledge about the biology and clinical features of ALT-positive neuroblastomas by enriching ALT tumors in the study cohort independent of *ATRX* mutation status. Using a multi-omics profiling approach of ALT-positive neuroblastomas, we provide evidence that this subgroup is clinically and molecularly distinct. Clinically, ALT is associated with a protracted course of disease and poor outcome. Molecularly, ALT-positive tumors feature mutated *ATRX* and/or reduced protein abundance, heterochromatic telomeric chromatin, a slow proliferative capacity, and frequent neo-telomere formation.

## Results

**Enrichment of ALT in relapsed disease.** Using the presence of circular partially single stranded extrachromosomal telomeric repeat sequences (C-Circles) as ALT marker[20], 9.2% of the neuroblastomas in a screening cohort of 720 tumors were classified as ALT-positive. C-Circle screening was further extended to a relapse cohort of tumors analyzed in the INFORM registry[21] ($n = 40$). Importantly, 47.5% of the INFORM tumors were ALT-activated, indicating a strong enrichment of this molecular subgroup in relapsed cases (Fig. 1a, Tables 1–2). Clinically, ALT was significantly associated with INSS stage 4, high-risk disease, and an older age at diagnosis. However, about one third of the ALT-positive neuroblastomas had localized disease and were stratified at diagnosis as low or intermediate risk using a standard risk classification system. In the screening cohort, none of the 83

tested 4 S stage tumors, a clinically recognizable subtype associated with spontaneous regression seen in infants below one year of age, were ALT-activated. Amplified *MYCN* and C-Circle presence were mutually exclusive in the screening cohort, but not in the INFORM cohort (Tables 1–2, Supplementary Data 1–2).

**Association of ALT with protracted clinical disease course and poor outcome.** Subsequently, neuroblastomas enriched for ALT were characterized using a multi-omics approach including high coverage whole genome sequencing ($n = 165$), RNA-sequencing ($n = 144$), whole proteome analysis ($n = 34$), and ChIP-sequencing ($n = 27$) (Fig. 1b, Supplementary Data 3, Methods) forming the discovery cohort. ALT tumors had a significantly higher telomere content (Fig. 2a). Telomere content and C-Circle intensity was comparable between ALT-positive tumors of the discovery and INFORM cohort (Supplementary Figs. 1–2). Overall, telomere content was positively correlated with C-Circle intensity ($r = 0.76$, $P < 2.2e^{-16}$, Spearman, Supplementary Fig. 1c). Minimal to no *TERT* mRNA expression was observed for ALT-positive neuroblastomas (Fig. 2b), resulting from epigenetic silencing of the *TERT* locus by H3K27me3 (Supplementary Fig. 3a). In accordance with low *TERT* mRNA expression, ALT-positive tumors exhibited low telomerase activity (Supplementary Fig. 3b). Moreover, ALT was associated with increased expression of the polyadenylated telomeric long noncoding RNA TERRA (Fig. 2c, Supplementary Fig. 3c). Patients with ALT-positive tumors had a similar poor event-free survival compared to patients with *MYCN*-amplified tumors. Although the overall survival time of patients with ALT-positive tumors was significantly longer, they had a dismal outcome in the long perspective (Fig. 2d). No difference in event-free or overall survival was observed between patients with ALT-positive tumors stratified into high-risk and those predicted as low/intermediate risk (Fig. 2e), indicating that the current risk stratification is underestimating the poor prognosis of ALT-positive tumors. ALT-positive neuroblastomas with a higher telomere content had a significantly shorter event-free survival as compared to ALT-positive tumors with relatively lower telomere content (Supplementary Fig. 3d). There was no clear trend in telomere content gain or loss between matching primary/relapse samples of the INFORM cohort (Supplementary Fig. 4). For 3 of 7 ALT-positive relapse tumors with a matching primary, ALT activity was only observed in the relapse tumor. Two of these cases were *MYCN* amplified in the primary disease period and heterogeneous ALT/MNA in the relapse. Heterogeneity could be confirmed by FISH analysis (*MYCN* and telomere probe) of one relapsed tumor (Supplementary Fig. 5a). For the other heterogeneous relapse tumor, we found evidence of an ALT-positive subclone in FISH analysis of the primary tumor (Supplementary Fig. 5b). Noteworthy, the discovery cohort also contains three heterogeneous ALT/MNA cases, which are *MYCN* amplified and have a very high telomere content, but were otherwise C-Circle negative (Fig. 2a, Supplementary Figs. 1a and 5c).

***ATRX* mutations in 55% of ALT-positive neuroblastomas.** The mutational landscape of ALT-positive neuroblastomas in the discovery cohort was very diverse with only few recurrent genetic events (Fig. 3a, Supplementary Fig. 6). ALT activation was mutually exclusive to *TERT* rearrangements (Fig. 3a). Loss-of-function mutations of *ATRX* were the most frequent events associated with ALT, but were only observed in 55% of ALT-positive tumors (Fig. 3a). *ATRX* mutations were mainly large deletions and single nucleotide variants (Fig. 3a–b, Supplementary Data 3–4). However, three patients had large intragenic *ATRX* duplications (Fig. 3a–b, Supplementary Data 3–4). Patient

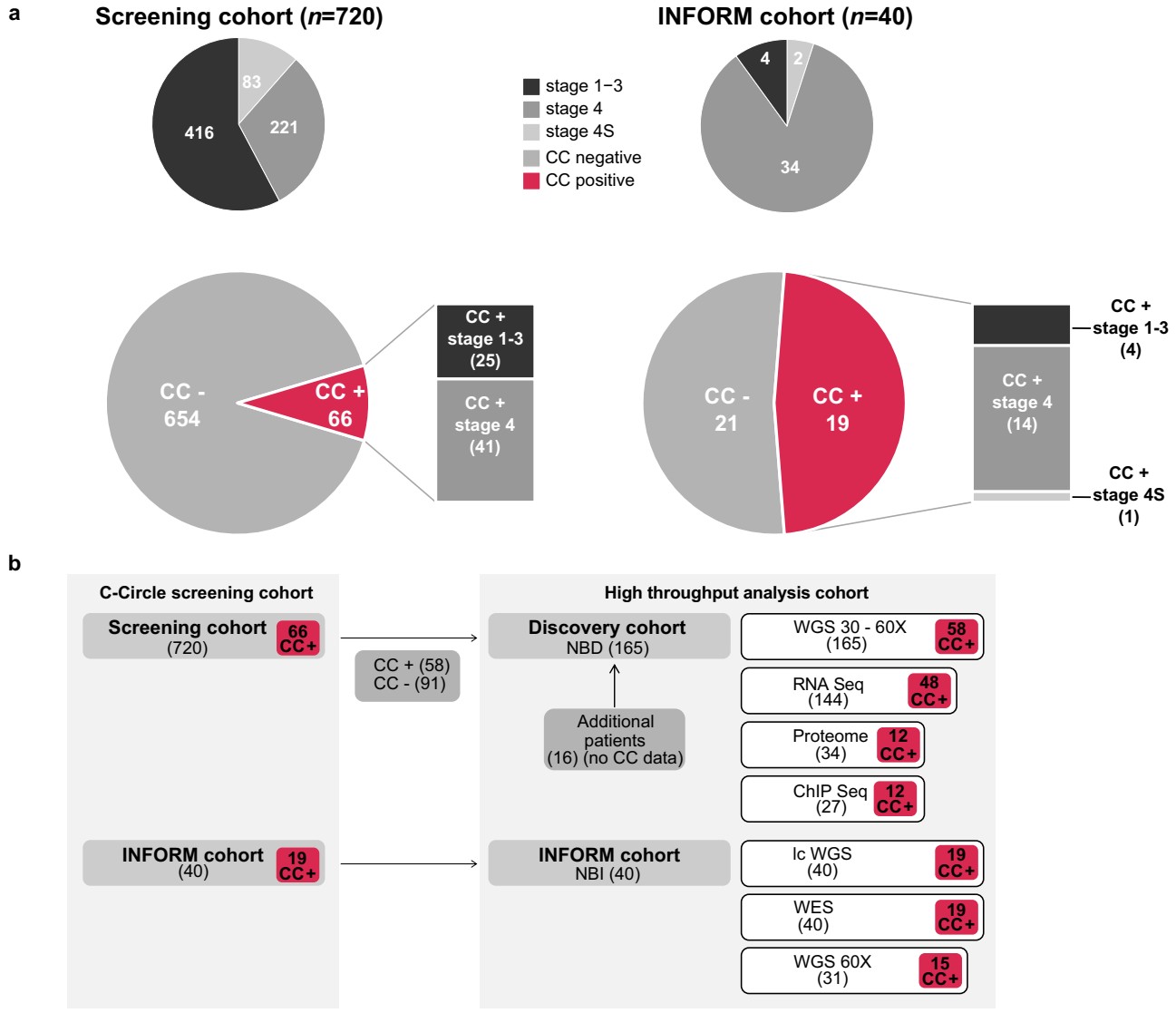

**Fig. 1 C-Circle screening of neuroblastomas. a** ALT frequency detected using C-Circle presence as a marker in the screening cohort ($n = 720$, left) and INFORM cohort ($n = 40$, right). Top pies illustrate the composition of the cohorts concerning INSS tumor stage. **b** Schematic overview of the analyzed patient cohorts used for C-Circle screening and high-throughput analysis using whole genome sequencing (WGS), RNA-sequencing, whole proteome analysis, and ChIP-sequencing. Sample numbers are given in brackets. Number of C-Circle positive (CC+) and C-Circle negative (CC-) patients in the sub-cohorts are indicated.

survival was not significantly different between *ATRX*-mutated and *ATRX* wild-type ALT-positive cases (Fig. 3c). Mutations in the ATRX complex members *DAXX* and *H3F3A* were extremely rare (Fig. 3a). Only one *ATRX* wild-type tumor exhibited a large inversion event affecting *DAXX*, associated with low *DAXX* mRNA expression (66.46 TPM; range discovery cohort 50-330 TPM). Another tumor had an A92S *H3F3A* mutation in addition to mutated *ATRX*. Somatic mutations in TP53 pathway genes (*TP53, CREBBP, ATM, ATR, CDKN2A*, and *MDM2*) were significantly enriched ($P = 0.01$) in ALT-positive tumors. ALT-positive tumors had the highest prevalence of *CDK4* amplifications (3/4) (Fig. 3a), associated with increased *CDK4* mRNA expression (Supplementary Fig. 7a), and showed higher *CCND1* mRNA expression as compared to telomerase-activated cases (Supplementary Fig. 7b). Overall, many tumors had mutations in one or another gene associated with telomere maintenance according to the TelNet database[22] (Fig. 3a, Supplementary Figs. 6 and 7c). Copy number loss of *POLD3* and *ATM* was frequently observed in ALT-positive tumors (Supplementary

Fig. 7c). However, both *POLD3* and *ATM* are located on chr11q and loss of these genes was associated with loss of chr11q, which is frequently observed in ALT-positive (Supplementary Fig. 8) and other non-*MYCN*-amplified neuroblastomas[23,24]. *Synaptic nuclear envelope protein 1* (*SYNE1*) was exclusively mutated in 7 of 60 ALT tumors (11.7%) (Fig. 3a, Supplementary Figs. 6, 9a). Furthermore, deletions in *receptor-type tyrosine-protein phosphatase delta* (*PTPRD*) were enriched in ALT-positive tumors ($P = 2.46e-05$; Fig. 3a, Supplementary Fig. 6). Structural variations affecting *PTPRD* were reported in neuroblastoma cohorts and low *PTPRD* expression was correlated with a poor prognosis[25,26]. However, structural variations of *PTPRD* were not correlated with reduced *PTPRD* mRNA expression in our discovery cohort (Supplementary Fig. 9b). Noteworthy, both genes are very large and thus the high mutation frequency in ALT neuroblastomas could also reflect the general increased mutational load found in older patients or a general increase in genomic instability (Supplementary Fig. 9c–e). The mutational landscape of relapsed neuroblastomas in the INFORM cohort was

**Table 1 Clinical features of C-Circle positive tumors in the screening cohort ($n = 720$)[a].**

| | Total cohort | | C-Circle positive | | C-Circle negative | | P value[g] |
|---|---|---|---|---|---|---|---|
| | No.[f] | %[f] | No.[f] | %[f] | No.[f] | %[f] | |
| Total | 720 | | 66 | 9.2% | 654 | 90.8% | |
| Stage 1-3[b] | 416 | 57.8% | 25 | 37.9% | 391 | 59.8% | 1.948E-06 |
| Stage 4[b] | 221 | 30.7% | 41 | 62.1% | 180 | 27.5% | |
| Stage 4S[b] | 83 | 11.5% | 0 | 0.0% | 83 | 12.7% | 0.000367 |
| LR/IR[c] | 449 | 65.9% | 24 | 36.9% | 425 | 69.0% | 6.944E-07 |
| HR[c] | 232 | 34.1% | 41 | 63.1% | 191 | 31.0% | |
| Male | 397 | 55.1% | 39 | 59.1% | 358 | 54.7% | 0.519 |
| Female | 323 | 44.9% | 27 | 40.9% | 296 | 45.3% | |
| Age[d] (mean in days) | 871 | | 2872 | | 669 | | <2E-16 |
| MYCN amp[e] | 114 | 16.2% | 0 | 0.0% | 114 | 17.9% | 5.883E-06 |
| MYCN normal[e] | 588 | 83.8% | 66 | 100.0% | 522 | 82.1% | |
| Ganglioneuroma | 11 | 1.5% | 0 | 0.0% | 11 | 1.7% | 0.6115 |

[a]$n$ refers to the number of tumors.
[b]Tumor staging was done according to the INSS criteria.
[c]Risk stratification was based on the German Neuroblastoma Study NB2004. Risk stratification was not available for 39 tumors.
[d]Age at diagnosis.
[e]MYCN status refers to MYCN status at diagnosis, which is either amplified (amp, $\geq$8 copies) or normal. MYCN status was not available or inconclusive for 18 tumors.
[f]All values are given in absolute number of tumors (No.) and relative (%) to the total number of tumors in each group (total cohort, C-Circle positive, C-Circle negative).
[g]P values were calculated using Fisher tests and two-sided Wilcoxon rank sum test (age).

**Table 2 Clinical features of C-Circle positive tumors in the INFORM cohort ($n = 40$)[a].**

| | Total cohort | | C-Circle positive | | C-Circle negative | | P value[g] |
|---|---|---|---|---|---|---|---|
| | No.[f] | %[f] | No.[f] | %[f] | No.[f] | %[f] | |
| Total | 40 | | 19 | 47.5% | 21 | 52.5% | |
| Stage 1-3[b] | 4 | 10.0% | 4 | 21.1% | 0 | 0.0% | 0.041 |
| Stage 4[b] | 34 | 85.0% | 14 | 73.7% | 20 | 95.2% | |
| Stage 4S[b] | 2 | 5.0% | 1 | 5.3% | 1 | 4.8% | 1.000 |
| LR/IR[c] | 7 | 17.5% | 5 | 26.3% | 2 | 9.5% | 0.226 |
| HR[c] | 33 | 82.5% | 14 | 73.7% | 19 | 90.5% | |
| Male | 32 | 80.0% | 13 | 68.4% | 19 | 90.5% | 0.120 |
| Female | 8 | 20.0% | 6 | 31.6% | 2 | 9.5% | |
| Age[d] (mean in days) | 1645 | | 2243 | | 1104 | | 0.003 |
| MYCN amp[e] | 11 | 27.5% | 2 | 10.5% | 9 | 42.9% | 0.029 |
| MYCN normal[e] | 28 | 70.0% | 17 | 89.5% | 10 | 47.6% | |

[a]$n$ refers to the number of tumors.
[b]Tumor staging was done according to the INSS criteria.
[c]Risk stratification was based on the German Neuroblastoma Study NB2004.
[d]Age at diagnosis.
[e]MYCN status refers to MYCN status at diagnosis, which is either amplified (amp, $\geq$8 copies) or normal. MYCN status was inconclusive for one tumor at diagnosis.
[f]All values are given in absolute number of tumors (No.) and relative (%) to the total number of tumors in each group (total cohort, C-Circle positive, C-Circle negative).
[g]P values were calculated using Fisher tests and two-sided Wilcoxon rank sum test (age).

comparable to the discovery cohort (Supplementary Fig. 10a, Supplementary Data 2, 3). In one matched primary relapse pair, an *ATRX* duplication event was retained in the relapse, further indicating that duplications can lead to nonfunctional ATRX (Supplementary Fig. 4). Notably, canonical activating RAS pathway mutations (*HRAS, NRAS, KRAS, BRAF, RAF1, CDK4, CCND1, NF1*) were significantly more frequent in relapsed ALT-positive neuroblastomas in the INFORM cohort compared to the discovery cohort ($P = 0.0013$, Supplementary Fig. 10, Fig. 3a), supporting a specific impact of RAS pathway mutations in relapsed ALT-positive tumors.

**Low ATRX protein abundance in ALT tumors.** Using a mass spectrometry based whole proteome approach ($n = 34$) quantifying on average 6891 proteins per tumor, we found 470 proteins significantly different in ALT-positive tumors compared to the other subgroups (nominal $P \leq 0.01$; fold change $\geq$2; Fig. 4a,

Supplementary Data 5). Enrichment analysis of significantly different mRNAs and proteins for each subgroup in turn highlighted both the overall agreement of transcriptomics and proteomics as well as the distinct effects only significantly present in one or the other omics-level (Fig. 4b, Supplementary Fig. 11a, Supplementary Data 6). The Ras family proteins HRAS and NRAS, comprised in the term "prenylation", were significantly upregulated in ALT-positive neuroblastomas (Fig. 4b, Supplementary Data 6). Among the top upregulated proteins in ALT-positive tumors were the cancer testis antigens P antigen family member 5 (PAGE5) and melanoma associated antigen 4 (MAGEA4), which may serve as potential targets for immunotherapies (Fig. 4a). Many pathways associated with proliferation were significantly downregulated on protein level in ALT-positive tumors compared to the telomerase-activated tumors including "DNA replication" and "chromosome". Notably, the classical proliferation marker MKI67[27], comprised in the term "chromosome", was one of the top significantly downregulated proteins in ALT-positive

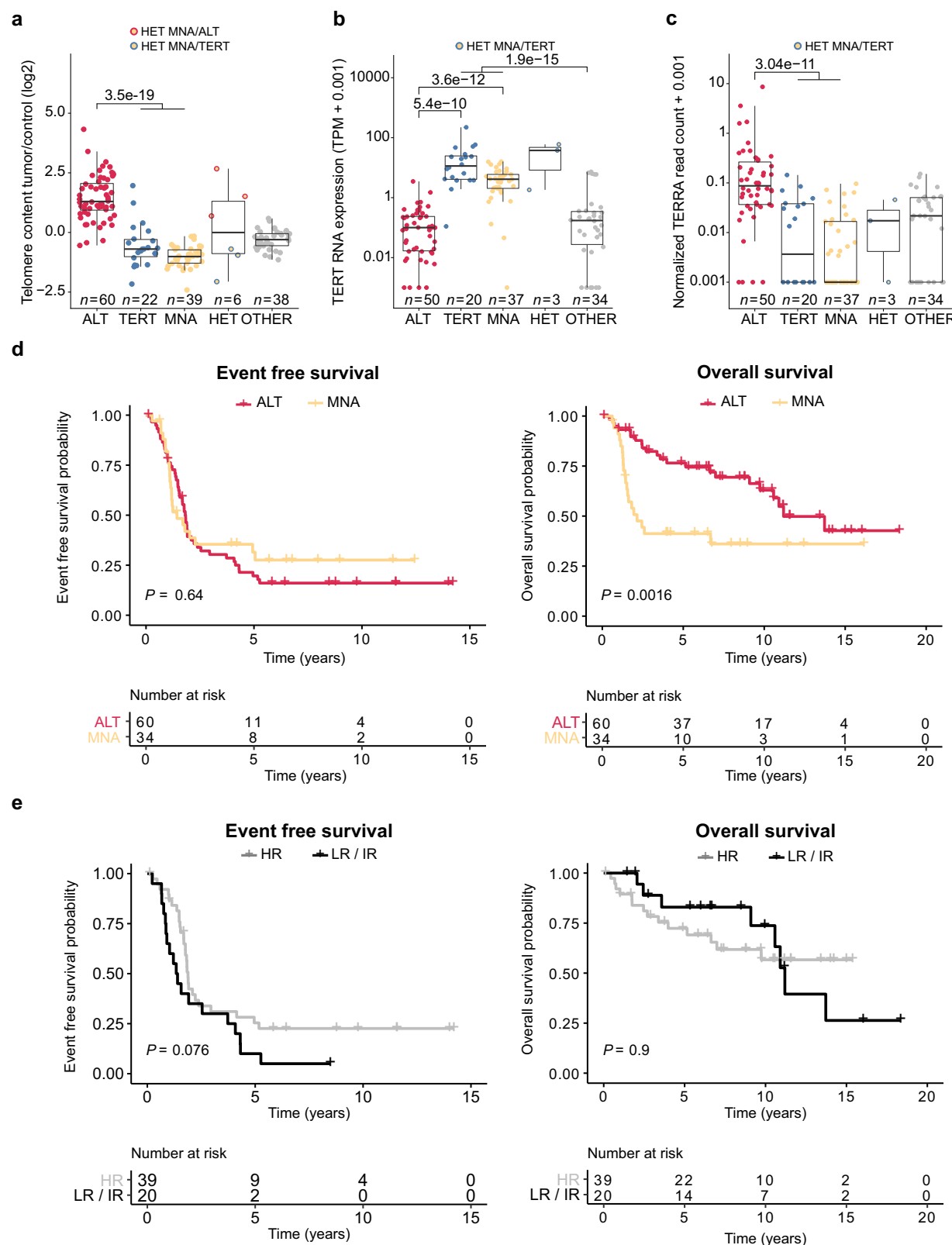

neuroblastomas (Fig. 4a–b, Supplementary Fig. 11b). The term DNA replication includes among others the proteins PCNA, MCM2, MCM3, MCM4, MCM5, and MCM7, which are also indicative of the proliferative capacity of a cell[28]. Furthermore, the term "bromodomain" was significantly reduced in ALT-positive tumors, while strongly upregulated in MNA tumors. Amplified *MYCN* is known to be associated with increased

occupancy of active promoter regions and enhancer invasion by MYCN and bromodomain proteins, leading to an increased transcriptional activity of many proliferation associated genes and downregulation of differentiation genes[29]. Moreover, ALT-positive tumors exhibit a significantly lower fraction of cycling cells compared to *MYCN*-amplified tumors (Supplementary Fig. 11c). Taken together, both protein expression data and cell

**Fig. 2 Characteristics and survival of ALT-positive neuroblastomas. a** Relative telomere content (tumor/control) of neuroblastoma tumors in the discovery cohort grouped according to the genetic evidence of an active telomere maintenance mechanism (see methods for details): ALT activation (ALT, red), telomerase activation by amplified *MYCN* (MNA, yellow), structural variation affecting *TERT* or *TERT* promoter mutation (TERT, blue), and patients having no genetic evidence of an activated telomere maintenance mechanism (OTHER, gray). **b** *TERT* mRNA expression in different subgroups. **c** Expression of the telomeric repeat-containing RNA TERRA in the different subgroups. **a–c** Boxplots indicate the median value (middle line), the 25th and 75th percentiles (box). The upper/lower whisker spans from the hinge to the largest/smallest value (values expanding a distance of 1.5 x inter quartile range are not considered). Dots represent individual tumors. *P* values were calculated with two-sided Wilcoxon rank sum tests. **d** Event-free and overall survival probability compared between ALT-positive ($n = 60$) and *MYCN*-amplified (MNA, $n = 34$) neuroblastomas. **e** Event-free and overall survival probability of patients with ALT-positive tumors classified as low or intermediate risk (LR/IR, $n = 20$) compared to patients with ALT-positive tumors classified as high-risk (HR, $n = 39$) using the NB2004 risk stratification. **d–e** *P* values were calculated using log rank tests. **a–e** *n* describes the number of analyzed tumors.

cycle analysis support a low proliferative capacity of ALT-positive neuroblastomas. 96 of the 470 significantly different proteins were annotated in the TelNet database to be associated with telomere maintenance[22] (Supplementary Data 5) ($P = 0.03$). The significantly downregulated proteins in ALT-positive neuroblastomas comprised ATRX, PCNA, EXO1, GNL3, KDM1A, RIF1, SCG2, TFAP2B, and GNL3L, all functionally linked to telomere maintenance[22]. Helicases and chromatin regulators, including ATRX, were less abundant at protein level in the ALT subgroup, but not at mRNA level (Fig. 4b). Importantly, ATRX itself was among the top candidates to exhibit lower protein abundance in ALT-positive tumors (Fig. 4a). Intriguingly, the reduced ATRX protein levels were independent of *ATRX* mRNA levels and mutation status (Fig. 5a). ATRX was among the most strongly downregulated proteins while the *ATRX* mRNA changes very little (Supplementary Fig. 11d), indicating that reduced ATRX protein abundance is a characteristic proteomic feature of all ALT-positive tumors (*ATRX* wild-type and mutated) that cannot be observed at the mRNA level. Analysis of exon-specific *ATRX* mRNA levels revealed that only the exons affected by deletion have reduced expression in the *ATRX*-deleted cases (Supplementary Fig. 12a). In summary, reduced ATRX protein level is a biomarker for ALT-positive neuroblastomas that is independent of mRNA levels and *ATRX* mutation. The decreased ATRX protein levels despite unchanged mRNA abundance could be due to reduced translation and/or increased degradation of ATRX in ALT-positive tumors. For example, it has been shown that unassembled subunits of multiprotein complexes (so-called orphans) are often degraded[30]. Downregulating one subunit can therefore induce degradation of other subunits, which partially explains divergence of protein- and mRNA-level changes in many biological contexts[31,32]. We therefore took a closer look at the ATRX binding partner DAXX. Interestingly, DAXX protein levels were specifically reduced in ALT-positive *ATRX* wild-type tumors (Fig. 5b). To investigate if reduced DAXX levels could explain ATRX downregulation at the protein level, we knocked-down *DAXX* in the ALT-negative neuroblastoma cell line NBL-S and observed that ATRX protein levels were indeed reduced after 96 h (Fig. 5c–d). Hence, the reduced ATRX protein levels in *ATRX* wild-type ALT positive tumors might result from downregulation of the DAXX protein. However, since *DAXX* mRNA levels do not differ significantly between tumors (Fig. 5b) and no recurrent mutation patterns in DAXX or ATRX/DAXX interacting proteins could be identified in ALT-positive *ATRX* wild-type tumors (Supplementary Figs. 12b, 13), the question what causes the downregulation of DAXX is still open. Hence, the mechanistic details behind ATRX/DAXX complex reduction remain to be uncovered.

**Heterochromatic telomeric chromatin in ALT neuroblastomas.** Many contradictory findings were reported on the telomeric chromatin landscape of ALT-positive cells[33–35], but data on

primary tumor samples was yet missing. Here, ALT-positive neuroblastoma tumors were found to be enriched for H3K9me3 at the telomeres (Fig. 6a). Overall levels of H3K9me3 at the telomeres of ALT-positive tumors were comparable to the H3K9me3 levels at SatII and SatIII alpha-satellite sequences (Supplementary Fig. 14a–b), which are known to be enriched for H3K9me3[35,36]. In line with a heterochromatic enrichment, protein expression of the demethylase KDM1A responsible for removing methyl groups from H3K9me3[37] was significantly lower in ALT-positive neuroblastomas compared to *TERT*-activated tumors (Fig. 6b). SETDB1 protein activity is required for H3K9me3 deposition at the telomeres, while the histone methyltransferases SUV39H1 and SUV39H2 catalyze H3K9 tri-methylation at heterochromatic sites[38–40]. Neither expression of *SETDB1* nor *SUV39H1/SUV39H2* could be associated with the increased telomeric H3K9me3 in ALT-positive neuroblastomas (Fig. 6c–d). H3K27ac at the telomeres was lower in ALT-positive tumors (Fig. 6a). However, this difference was not telomere specific and similarly observed at SatII and SatIII sequences indicating a general difference in H3K27ac (Supplementary Fig. 14c). Furthermore, ALT-positive tumors displayed reduced telomeric H3K27me3 (Fig. 6a, Supplementary Fig. 14d) and low protein and mRNA expression of the PRC2 component EZH2, responsible for H3K27me3 deposition (Fig. 6e). Low H3K36me3 was observed at the telomeres and satellite sequences of both ALT-positive and ALT-negative neuroblastomas (Fig. 6a, Supplementary Fig. 14e). Together, this indicates epigenetic dysregulation of repressive marks at the telomeres in ALT-positive tumors with excess of H3K9me3 and reduction of H3K27me3.

**Increased frequency of telomeric repeat loci in ALT tumors.** ALT-positive neuroblastomas exhibited an increased frequency of telomeric repeat loci compared to the other groups (Fig. 7a), with ALT-positive tumors displaying up to 19 such events per tumor. The telomeric repeat loci frequency was significantly higher in *ATRX*-mutated than *ATRX* wild-type ALT tumors. Telomeric repeat loci are characterized by a chromosomal junction site and the presence of telomeric repeat sequences either upstream or downstream of this junction site (one-sided) (Fig. 7b). In rare cases, two telomeric repeat loci in close proximity (10 kb) with opposite orientation can form a two-sided event (Supplementary Fig. 15a). In our discovery cohort, only five such two-sided events were observed (Supplementary Fig. 15b). Two-sided events can result either from two separate events in close proximity or from telomere sequences that are spanning the region between the two junction sites forming a true insertion. Evidence for a true insertion, meaning that mates of a read pair matched to both sides of the insertion could only be identified for two of these events (Supplementary Table 1). The presence of more than three telomeric repeat loci in a tumor was found to be associated with a significantly reduced overall survival in ALT-positive neuroblastomas (Fig. 7c). Hotspot regions for recurrent telomeric

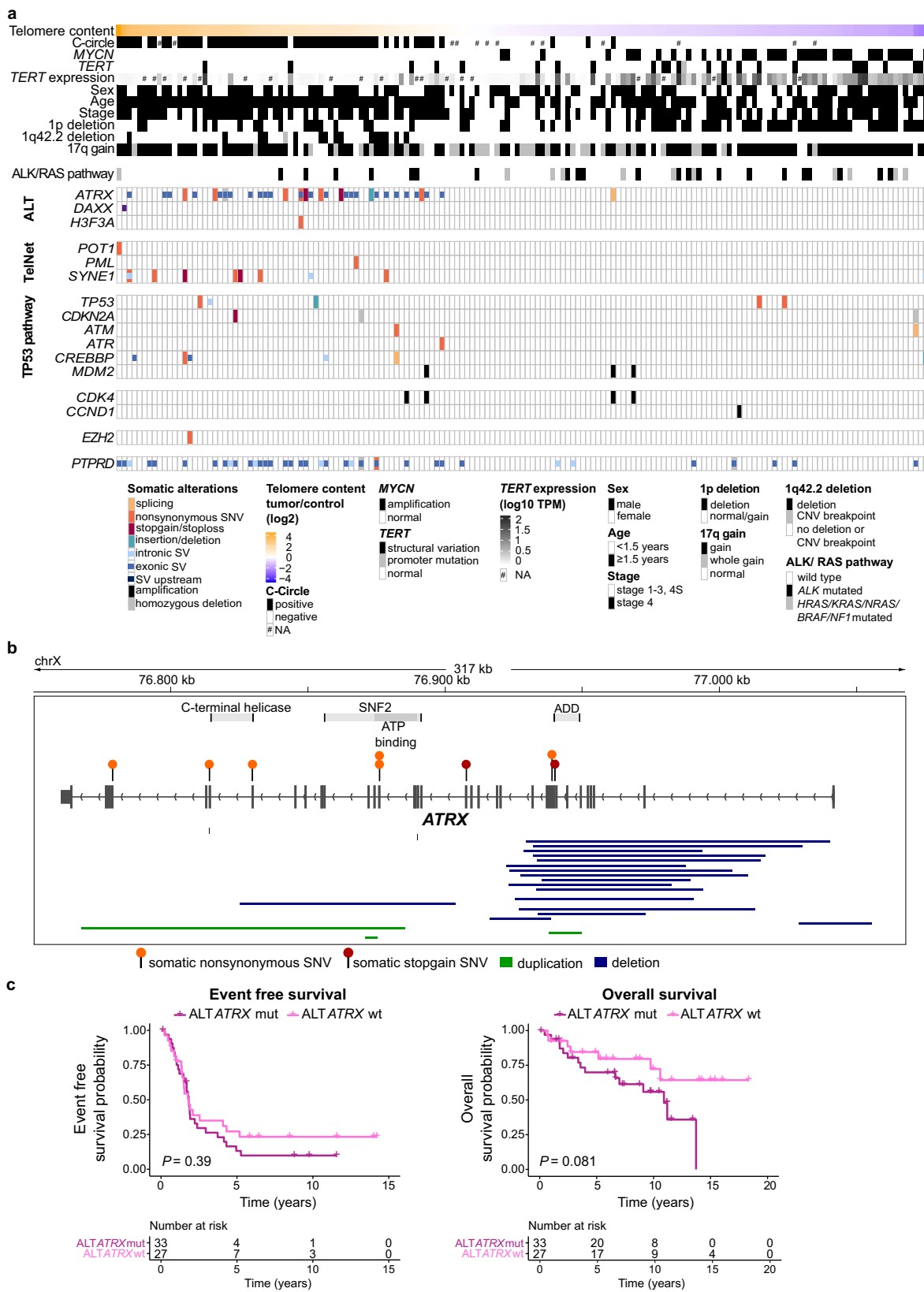

repeat loci were found on chromosomes 1q42.2, 18q23, and 19q13.43 (Fig. 7d). The increased occurrence of telomeric repeat loci in ALT-positive neuroblastomas and their enrichment on 1q42.2 was confirmed in the INFORM dataset (Supplementary Fig. 15c). 80.3% (212/264) of all t-type (TTAGGG) containing telomeric repeat loci showed at least 1 bp microhomology to the

canonical telomeric repeat sequence, which is more than one would expect by chance. This enhanced microhomology at the junction sites of telomeric repeat loci indicates that telomeric sequences were added via a microhomology-dependent mechanism (Supplementary Fig. 15d-f). In general, 55.3% of all telomeric repeat loci were associated with a chromosomal copy

**Fig. 3 Mutational landscape of ALT-positive neuroblastomas. a** Somatic alterations in the discovery cohort. Samples ordered based on relative telomere content. Exonic single nucleotide variations (SNVs), exonic small insertions and deletions (INDELs) and structural variations in exonic and intronic regions (SVs) in selected ALT-associated genes, genes associated with telomere biology according to the TelNet database[22] [https://malone2.bioquant.uni-heidelberg.de/fmi/webd/TelNet] and *TP53* pathway genes are illustrated. Telomere maintenance features, clinical features, chromosomal aberrations, and *ALK/RAS* pathway mutations (*HRAS, NRAS, KRAS, BRAF, NF1* or *ALK*) are given in the top panel. **b** Exact genetic location of mutations affecting the *ATRX* gene. Bars represent structural variations (deletion, duplication) and small deletions, while SNVs are illustrated as lollipops and colored by the SNV type. **c** Event-free and overall survival rate of patients with an *ATRX*-mutated ALT-positive neuroblastoma (n = 33) compared to patients with an *ATRX* wild-type ALT-positive neuroblastoma (n = 27). P values were calculated using a log rank test. n describes the number of analyzed tumors.

number loss of the neighboring segment (Supplementary Fig. 15g). A smaller fraction of loci (4.9%) was associated with a copy number gain. However, 30.7% of the telomeric repeat loci were copy number neutral. Some of these copy number neutral events were associated with one or more structural variations, but for 77.2% of the copy number neutral telomeric repeat loci no other event could be detected (Supplementary Fig. 15g).

**1q42.2 telomere repeat locus coincides with 1q42-1qter deletions**. Telomeric repeat loci contributing to the hotspot on chr1q42.2 (230.370 mb – 231.433 mb, Fig. 7e) frequently coincided with loss of the adjacent segment of chr1q (Fig. 7f). Telomeric repeat loci at 1q42.2 showed a high degree of microhomology to the telomeric TTAGGG repeats (Supplementary Figs. 15e, 16a). Loss of chromosome 1q42.2-1qter was also observed in some tumors without the occurrence of telomeric repeats at this site. However, 1q42.2-1qter deletions were exclusively seen in ALT-positive tumors, most of which had an *ATRX* mutation (Supplementary Figs. 16b, 8). RNA expression of most genes in the deleted 1q region was lower in 1q42.2-1qter deleted compared to non-deleted tumors (Fig. 7g). Among the significantly reduced genes in 1q42.2-1qter deleted tumors were the genes *ARID4B, EXO1, FH, CNST, ADSS, HNRNPU*, and *RYR2*, which are known to be associated with telomere maintenance and biology[22]. Of note, expression of the 5′–3′ exonuclease, *EXO1*, located on chromosome 1q43 and involved in DNA double strand break repair[41,42], telomere replication, and t-loop formation[43], was not only reduced in 1q42.2-deleted tumors but generally reduced in ALT-positive neuroblastomas (Supplementary Fig. 16c).

## Discussion
In summary, this in-depth analysis revealed that ALT-positive neuroblastomas are biologically and clinically distinct from tumors with telomerase activation.

ALT in neuroblastoma was associated with characteristic ALT features, including high telomere content, low *TERT* expression, and increased TERRA expression, which is in line with previous findings on ALT-positive tumors[44,45]. *ATRX* mutations were identified in 55% of the ALT-positive neuroblastomas. In line with this observation, an *ATRX* mutation frequency of 60% was reported in a parallel study[46]. The most frequent type of *ATRX* mutations were large deletions. However, we could also identify focal intragenic *ATRX* duplication events. Focal duplication of *ATRX* was also observed in a matching primary/relapse pair indicating that these events are leading to loss of ATRX. Similar events resulted in loss of ATRX function in patients with the genetic disorder alpha thalassemia X-linked intellectual disability syndrome (ATRX-syndrome)[47,48] further supporting a loss of ATRX function by partial duplications.

Integrating proteomic profiling revealed reduced ATRX/DAXX protein complex abundance as a recurrent event in ALT-positive neuroblastoma, which could often not be explained by mutations in these genes. The observation that all five ALT-positive tumors with wild-type *ATRX* depicted reduced DAXX protein levels is intriguing. It is tempting to speculate that the ALT phenotype always results from loss of the ATRX/DAXX complex activity, which is caused either by *ATRX* mutations or by reduced ATRX/DAXX protein levels. Future studies will show if this is indeed the case in neuroblastoma and/or possibly other ALT-positive cancers. Nevertheless, we could identify a subgroup of ALT-positive neuroblastomas with wild-type *ATRX*, but low ATRX protein abundance. Reduced ATRX protein levels in ALT *ATRX* wild-type neuroblastoma could be explained by the reduced DAXX protein levels, which we specifically observed in this subgroup of tumors. In this scenario, reduced DAXX protein levels impair assembly of the ATRX/DAXX complex, which then causes degradation of orphan ATRX protein molecules. Consistent with this idea, we observed that knocking down *DAXX* reduced ATRX protein levels in neuroblastoma cells. However, the cause of reduced DAXX protein levels is not yet clear, especially since mRNA levels do not change significantly. Reduced DAXX protein levels in ALT-positive tumors may result from posttranscriptional events. It is known that DAXX is regulated via various posttranslational modifications including phosphorylation, SUMOylation, and ubiquitination[49]. Irrespective of the mechanistic details involved, our study highlights that proteomic data is closer to phenotypes than transcriptomic data, which is especially valuable in a clinical context[32].

A recent report stated that *ATRX* mutation and amplified *MYCN* are mutually exclusive[50]. C-Circle screening of a cohort of 720 neuroblastomas also found a mutual exclusivity of C-Circle presence and amplified *MYCN*. However, in the INFORM cohort we identified two cases with amplified *MYCN* and a positive C-Circle signal. Additionally, in our discovery cohort we classified three cases as ALT/MNA heterogeneous. These cases were negative in the C-Circle assay, but exhibited a very high telomere content. FISH analysis of some of these cases could also confirm that amplified *MYCN* and ultra-bright telomere spots are present in the same tumor (Supplementary Fig. 5). No *ATRX* mutations were detected in the MNA/ALT heterogeneous tumors (for NBI26 only lcWGS and WES data was available). Taken together, heterogeneous tumors with amplified *MYCN* and presence of ALT markers seem to occur, but are very rare. Further analysis and larger sets of these cases would be necessary to draw meaningful conclusions on the clonality of these events and the activity of the respective telomere maintenance mechanism.

ChIP-sequencing analysis of ALT-positive neuroblastoma tumors showed that telomeres as well as SatII and SatIII sequences show lower H3K27ac compared to ALT-negative tumors. Furthermore, proteins comprised in the term "bromodomain", including EP300 responsible for H3K27ac, were reduced in ALT tumors and enhanced in telomerase-activated tumors. Moreover, the telomeres of ALT-positive neuroblastoma tumors were enriched for H3K9me3. These observations confirm the results of a previous study on cell lines[35] that only the telomeres of ALT-positive tumors are enriched for H3K9me3. In addition, we observed low H3K27me3 at the telomeres of ALT cases, which goes in line with a significantly

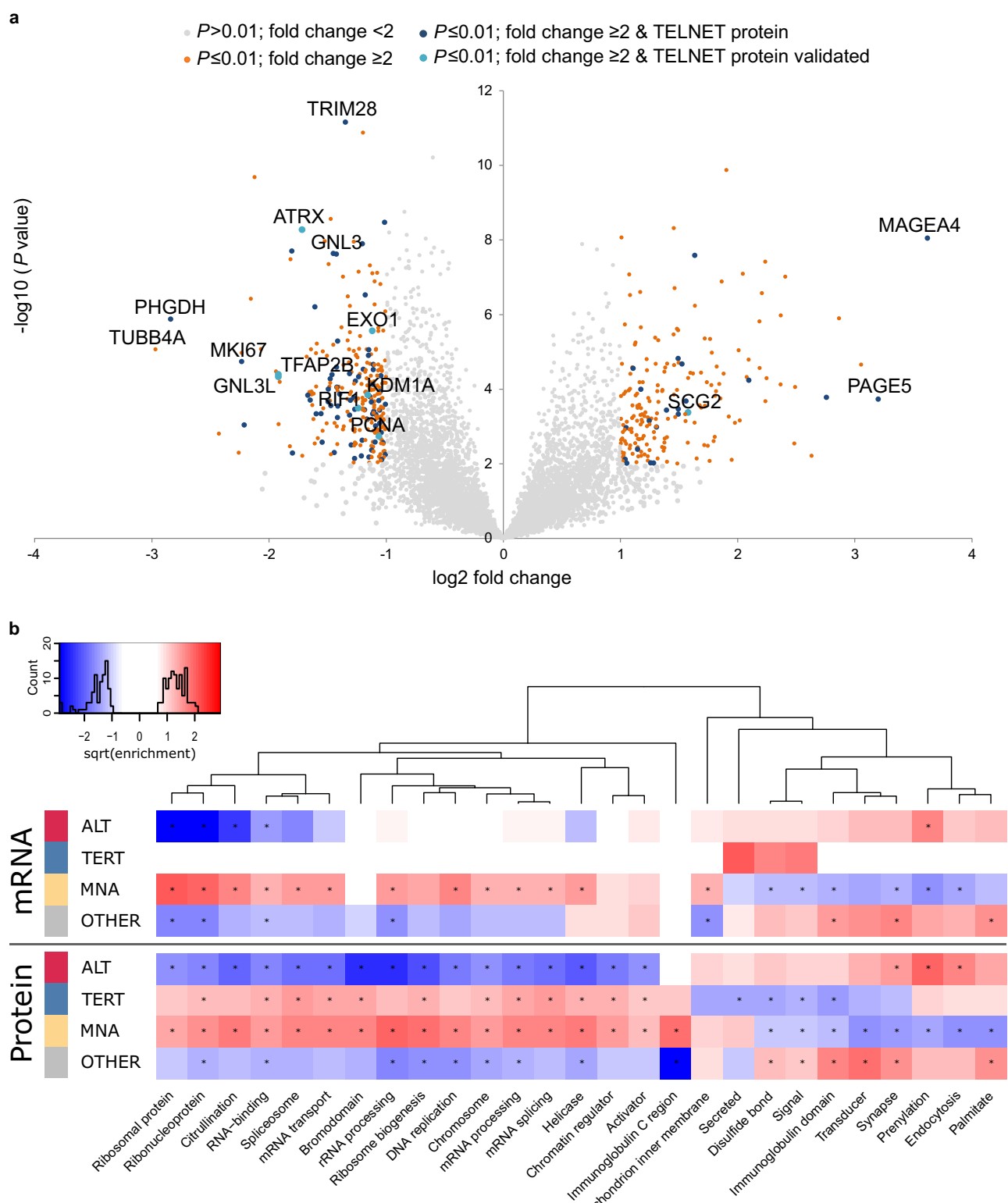

**Fig. 4 Proteomic analysis of ALT-positive neuroblastomas. a** Label free quantification (LFQ) analysis comparing ALT tumors with MNA and TERT tumors as well as tumors without a genetic telomere maintenance mechanism (OTHER). *P* values were calculated using a two-sided Student's *t*-test. Colors indicate *P* value and fold change categories as well as the association to telomere biology based on the TelNet database[22] [https://malone2.bioquant.uni-heidelberg.de/fmi/webd/TelNet]. **b** Differences between neuroblastoma subtypes (ALT, TERT, MNA, OTHER). UniProt keywords ([https://www.uniprot.org/keywords/]; retrieved on 2016-11-02) with significant enrichments (BH adjusted *P* < 0.000005, at least 2.5-fold enriched, at least 3 protein/mRNAs per keyword) in at least one group with significantly more or less abundant mRNA or proteins comparing each group to all other samples in turn (BH adjusted *P* < 0.05, * marks significant enrichment). Positive and negative sqrt enrichment values for enrichments in more and less abundant mRNA or protein, respectively. Enrichment *P* values were calculated by hypergeometric test and multiple-testing adjusted by Benjamini–Hochberg method. Exact *P* values and individual proteins contributing to Uniprot keywords are given in Supplementary Data 6.

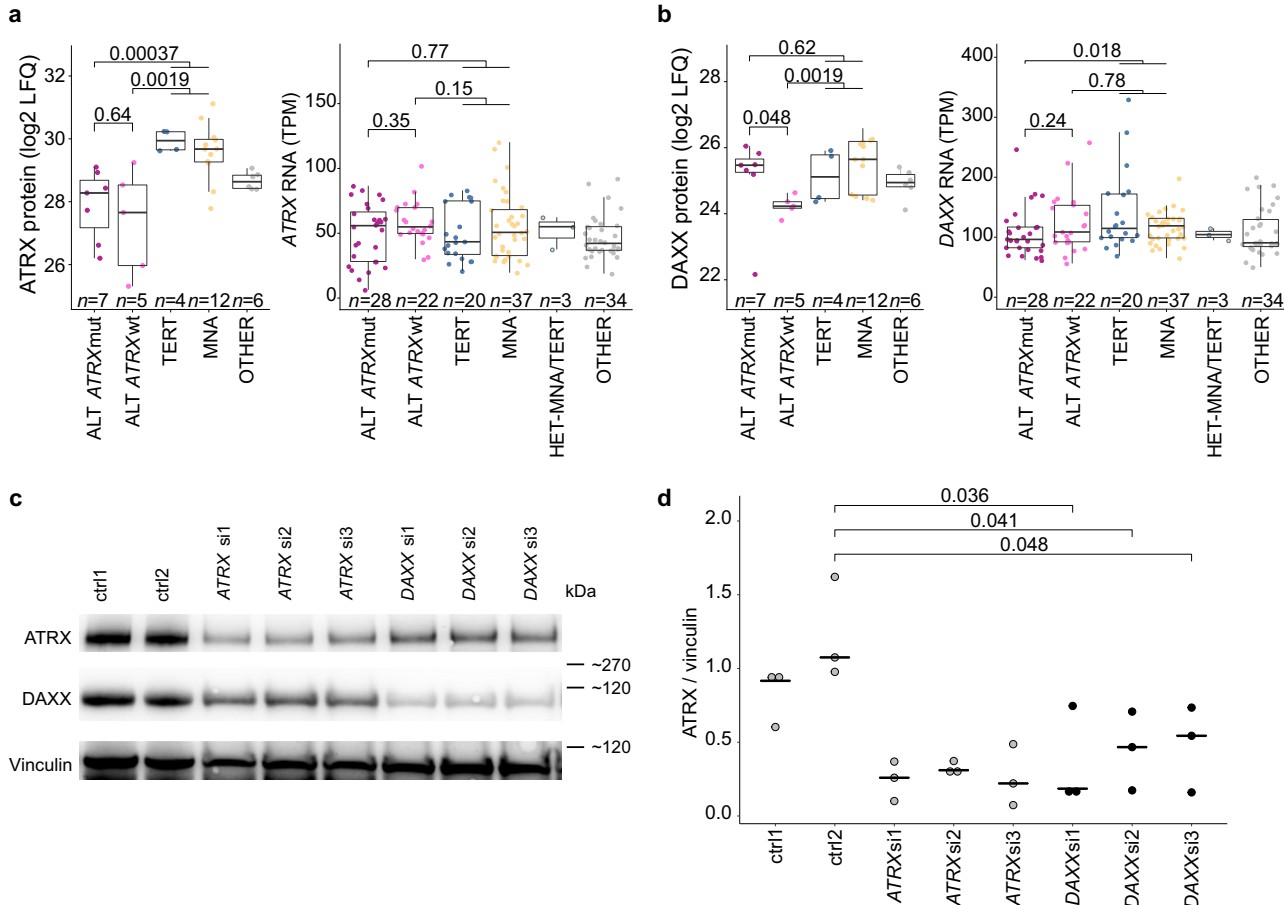

**Fig. 5 ATRX/DAXX complex abundance. a** ATRX protein and *ATRX* mRNA expression in ALT *ATRX*-mutated (dark pink) and wild-type (light pink) tumors compared to TERT (blue), MNA (yellow), and OTHER (gray). **b** Boxplots of DAXX protein and *DAXX* mRNA expression in ALT *ATRX*-mutated (dark pink) and wild-type (light pink) tumors compared to TERT (blue), MNA (yellow), and OTHER (gray). **a**, **b** Boxplots indicate the median value (middle line) and the 25th and 75th percentiles (box). The upper/lower whisker spans from the hinge to the largest/smallest value (values expanding a distance of 1.5 x inter quartile range are not considered). Dots represent individual tumors. *P* values in boxplots were calculated with two-sided Wilcoxon rank sum tests. *n* describes the number of analyzed tumors. **c**, **d** ATRX protein expression in NBL-S cells (with moderate *MYCN* expression) after 96 h treatment with siRNAs against *DAXX* (black). *ATRX* siRNAs as well as two negative control siRNAs were used as controls (gray). Exemplary western blot (**c**) and quantification of three biological replicates (**d**) is shown. *P* values calculated using a two-sided Student's *t*-test. Horizontal line shows median value. Dots represent biological replicates. Uncut images of all biological replicates are shown in the source data file.

lower protein and mRNA expression of *EZH2* in ALT neuroblastomas compared to telomerase-activated tumors.

ALT-positive neuroblastoma tumors exhibited a high rate of telomeric repeat loci. Since telomeric repeat loci were characterized by telomeric repeat sequences either upstream or downstream of a non-telomeric junction site, these events cannot have occurred from breakage of interstitial telomeric sequences (ITS). However, telomeric repeat loci frequently overlapped with breakpoints of copy number changes or structural variations. Because terminal chromosomal breaks are in need of telomeric repeats to protect the newly formed ends from degradation[51–53], we propose that ALT-positive tumors are capable of adding telomeric repeat sequences to open ends of chromosomal breaks forming neo-telomeres (Fig. 8a). The high degree of microhomology to the telomeric repeat sequence at the junction sites indicates that telomeric repeats are added via a microhomology-dependent process like microhomology-mediated end-joining or non-homologous end-joining[54]. Further, we propose that the presence of microhomology at an open chromosomal break determines if telomeric sequences can be added to this site. Chromosomal loss of certain fragments might present a selection advantage leading to a selection of cells harboring the neo-telomere. We also identified a subset of copy

number neutral telomeric repeat loci with no associated structural variation. This might be due to the fact that these events are sub-clonal and were thus not detected by the CNV/SV calling algorithm. Moreover, the detection limit of the used copy number algorithm is 50 kb and thus smaller copy number changes cannot be detected. Two-sided events, defined as two telomeric repeat loci in a 10 kb window with opposite orientation of the telomeric repeats, were very rare. These events may represent insertions of telomeric repeat sequences similar to previously described events[55]. Only two of five two-sided events exhibited evidence of a true insertion by mates of a read pair mapping to both sides of the insertion. However, for large insertions the used short read sequencing prevents the detection of supporting reads. Alternatively, two-sided events may result from neo-telomere formation on both sides of a breakpoint (Fig. 8b). A recurrent hotspot for telomeric repeat loci was identified on chr1q42, which was associated with 1q42.2-1qter loss forming a neo-telomere on chr1q42.2. Telomeric repeat loci were also associated with ALT and overlapped with chromosomal breakpoints in a pan-cancer cohort, but no enrichment was seen for telomeric repeat loci on 1q42.2[45], indicating that neo-telomere formation on chromosome 1q42.2 is neuroblastoma-specific. 1q42.2-1qter deletions retained in relapsed ALT cases indicate a selective advantage

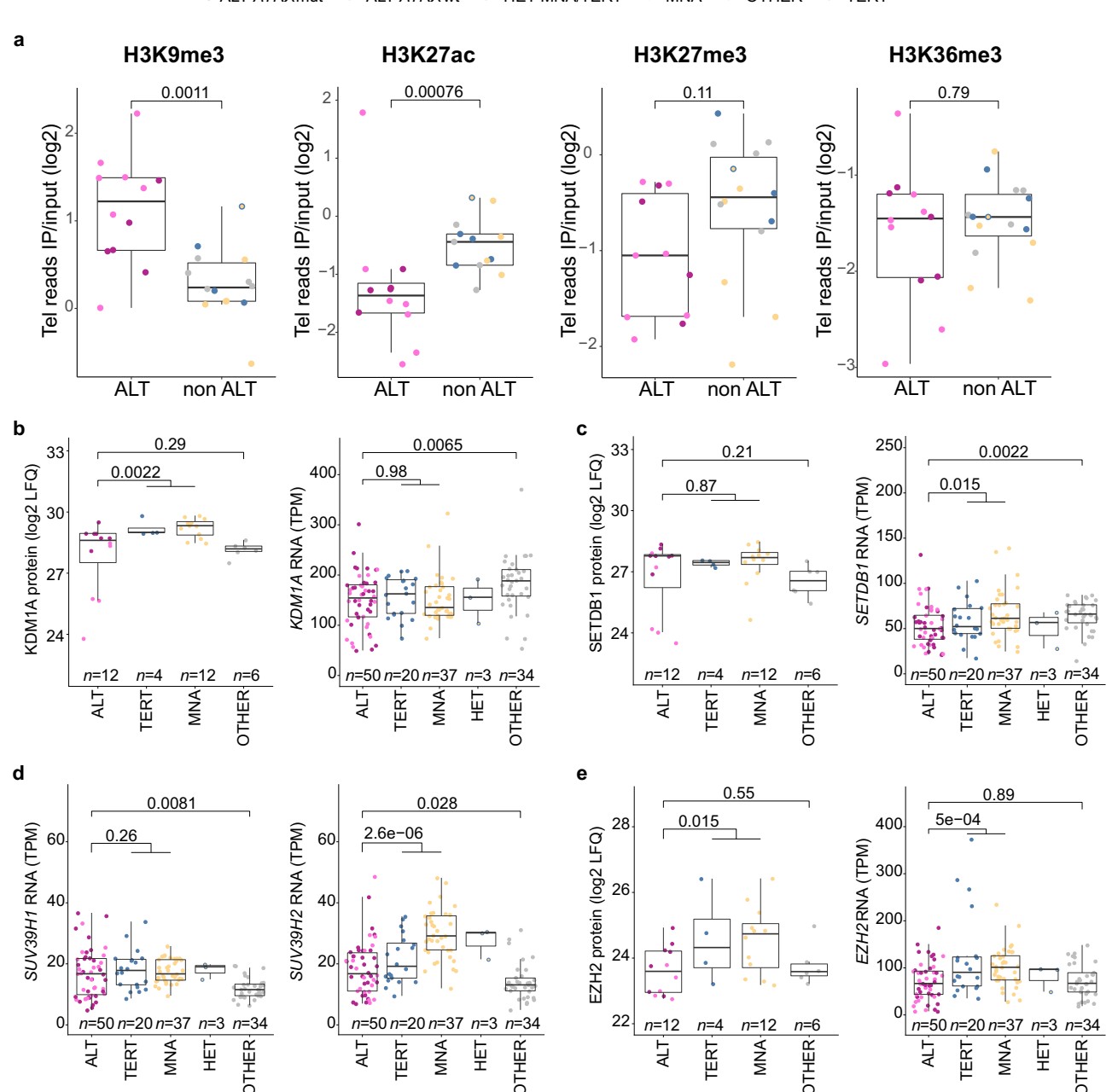

**Fig. 6 Telomeric chromatin state of ALT-positive neuroblastomas. a** Telomere repeats in ChIP of H3K9me3 ($n = 26$), H3K27ac ($n = 25$), H3K27me3 ($n = 25$) and H3K36me3 ($n = 27$) relative to telomere repeats in input of ALT-positive and ALT-negative tumors. **b–e** Protein and mRNA expression of *KDM1A* (**b**), *SETDB1* (**c**), *SUV39H1/H2* (**d**, only RNA), and *EZH2* (**e**) in ALT tumors compared to tumors form the other groups (TERT, MNA, OTHER). **a–e** Colors indicate subgrouping based on presence and type of telomere maintenance mechanism (see legend). Boxplots indicate the median value (middle line) and the 25th and 75th percentiles (box). The upper/lower whisker spans from the hinge to the largest/smallest value (values expanding a distance of 1.5 x inter quartile range are not considered). Dots represent individual tumors. *P* values were calculated using a two-sided Wilcoxon rank sum test. *n* describes the number of analyzed tumors.

of the deletion. Furthermore, chr1q42.2-1qter deletions were associated with reduced expression of candidate genes involved in telomere biology, including *EXO1*. Interestingly, EXO1 expression is not only reduced in ALT-positive tumors with 1q42.2-1qter deletions, but in ALT tumors in general. This indicates a potential role of EXO1 as an ALT suppressor that may be silenced by loss of 1q42.2 or other means.

ALT activity was associated with unfavorable outcome and a protracted clinical course of disease, which further substantiates previous observations[56]. Furthermore, ALT was enriched in a

population-based clinical sequencing study cohort for relapsed neuroblastomas. Multiple factors may contribute to this enrichment in the INFORM cohort. The INFORM inclusion criteria might contribute to an underrepresentation of the most aggressive neuroblastomas, which are most likely *MYCN*-amplified or *TERT*-rearranged, since patients are required to have sufficient general condition and a life expectancy of at least three months to be included in the registry trial. Furthermore, we could show risk underestimation of a substantial proportion of ALT-positive tumors, which was initially

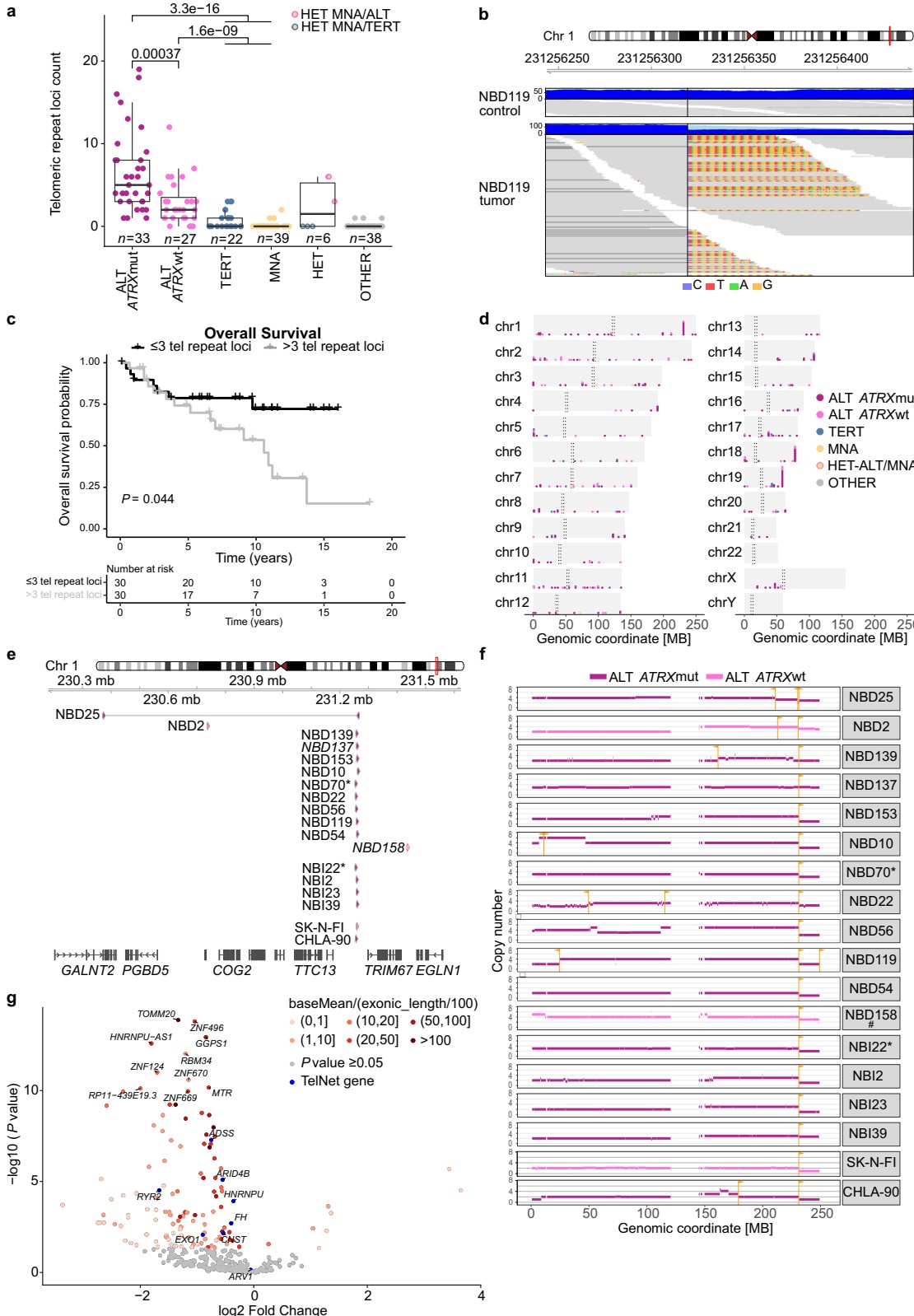

diagnosed as low or intermediate risk, but nevertheless had poor outcome. The current standard treatment regimens targeting strongly proliferating tumors are probably not suited to treat slowly growing ALT tumors, which might lead to a selection and enrichment of ALT-positive cases or subclones in the relapse setting. Thus, ALT activation might present a

therapy resistance mechanism. Analysis of matching primary/relapse pairs revealed a gain of ALT or outgrowth of an ALT-positive subclone in the relapse tumor. Taken together, a revised therapy concept based on a better molecular understanding of this subtype and considering ALT-specific vulnerabilities is urgently needed for these children.

**Fig. 7 Telomeric repeat loci. a** Telomeric repeat loci frequency in *ATRX*-mutated (dark pink) and ATRX wild-type (light pink) ALT-positive tumors compared to TERT (blue), MNA (MNA), and tumors without genetic evidence of an activated telomere maintenance mechanism (OTHER, gray) in the discovery cohort. Boxplots indicate the median value (middle line) and the 25th and 75th percentiles (box). The upper/lower whisker spans from the hinge to the largest/smallest value (values expanding a distance of 1.5 x inter quartile range are not considered). Dots represent individual tumors. *n* describes the number of analyzed tumors. *P* values were calculated with two-sided Wilcoxon rank sum tests. **b** Example of one telomeric repeat locus in tumor NBD119. Dark blue color indicates sequencing coverage and light blue color illustrates clipped sequences. Reads are shown in medium gray and clipped bases are color-coded. Non-telomeric ends of a discordant read pair are labeled in dark gray. **c** Overall survival probability of ALT-positive tumors harboring more than three telomeric repeat loci compared to ALT-positive tumors with three or less than three events. *P* value calculated using a log rank test. **d** Chromosomal location of telomeric repeat loci in the discovery cohort. **e** Exact genomic location of events contributing to the hotspot on chromosome 1q42.2 in the discovery and INFORM cohort and two ALT-positive cell lines. The 1q42.2 events of NBD158 and NBD137 ware added manually (see Methods). **f** Chr1 copy number profiles of tumors with a telomeric repeat locus on chr1q42.2. Junction sites and orientation of telomeric repeats are indicated in orange. Color-coding according to subgroup (# HET ALT/MNA). **g** Differential mRNA expression of genes located in the chr1q42.2-1qter deleted region comparing 1q42-1qter deleted to non-deleted tumors. Genes are colored by their mean expression normalized by exonic length or by their presence in the TelNet database[22]. *) Matched primary relapsed pair.

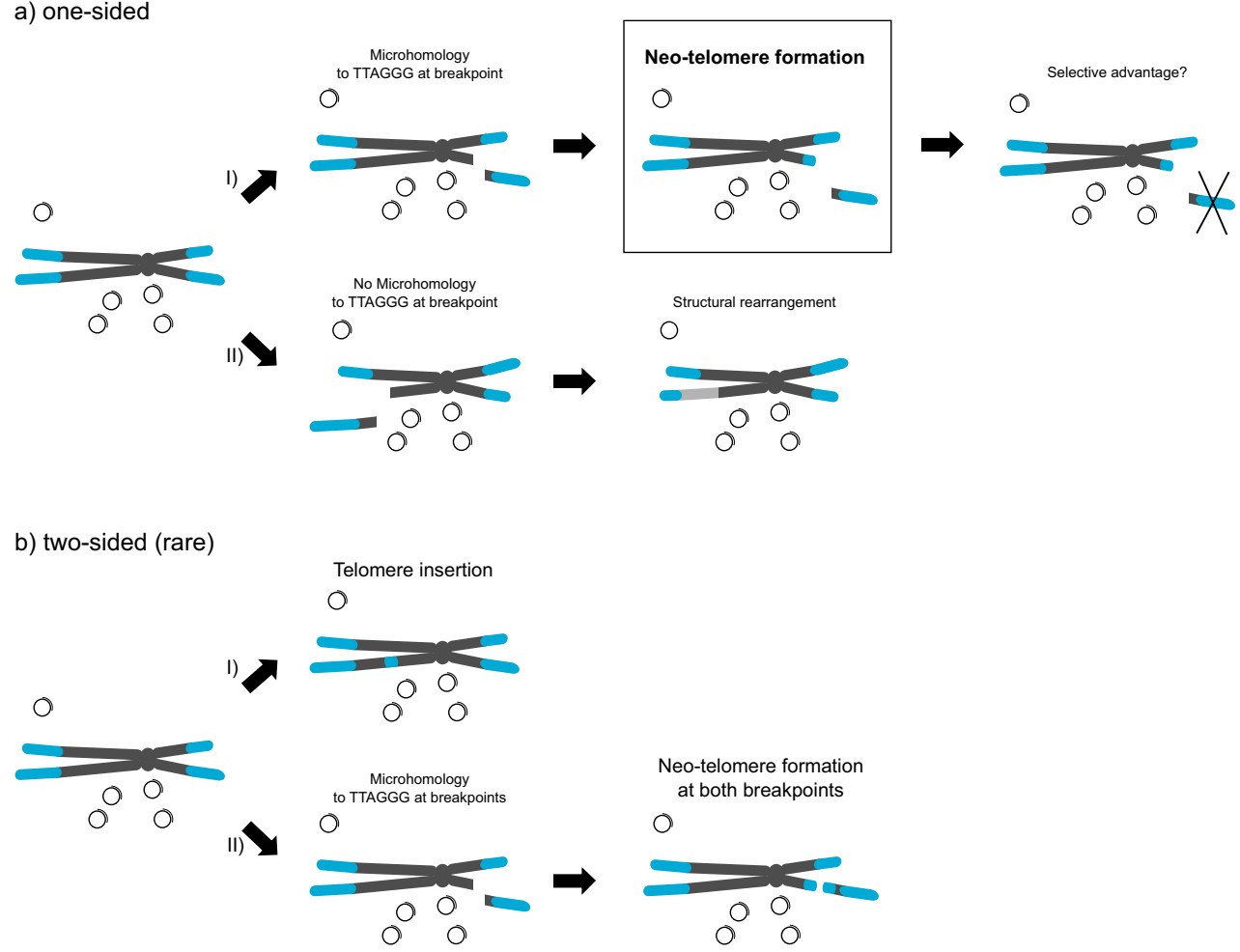

**Fig. 8 Model of neo-telomere formation.** Graphical abstract illustrating the hypothesis of neo-telomere formation in ALT-positive neuroblastomas. **a** The majority of telomeric repeat loci are one-sided. We hypothesize that ALT-positive cells are able to add telomeric sequences to open chromosomal breaks to protect them from degradation. **I** Microhomology to TTAGGG favors the formation of a neo-telomere at the open chromosomal break. Loss of some chromosomal regions might present a selection advantage for the cell and cells with a neo-telomere at the chromosomal breakpoint are selected. **II** Without microhomology, structural rearrangements involving other chromosomal arms (light gray) can represent an alternative route of protecting open chromosomal breaks. **b** Rare two-sided events can result from insertion of telomeric sequences **I** or from the formation of neo-telomeres on both sides of the breakpoint **II**. Small circles represent C-Circles. Chromosomes are shown in gray and telomeric sequences in blue.

## Methods

**Patient cohorts and tumor samples.** We retrospectively analyzed cohorts of primary and relapsed neuroblastoma tumors. ALT-activity was determined using C-Circle assays for 720 neuroblastoma tumors (screening cohort, Supplementary Data 1). High coverage whole genome sequencing (WGS) data was obtained for a subset of 165 tumors (discovery cohort). RNA-sequencing (*n* = 144), whole

proteome analysis (*n* = 34), and ChIP-sequencing (*n* = 27) was done for subsets of the discovery cohort. A summary of the clinical information and available data for every patient in the discovery cohort is given in Supplementary Data 3. Patients were enrolled in the neuroblastoma clinical trials (NB2004-HR/NCT03042429, GPOH-NB2004/NCT00410631, GER-GPOH-NB97/NCT00017225, GER-NB95-S/NCT00002803, GER-NB90/NCT00002802, NB2016-registry) of the Society for

Pediatric Oncology and Hematology (GPOH) between 1991 and 2018. All trials were approved by the Ethics Committee of the Medical Faculty, University of Cologne and collection and use of all tumor tissue material was approved. Tumors of nine Austrian neuroblastoma patients were analyzed as part of the screening and discovery cohort. Tumor and control DNA/RNA was provided by Prof. Dr. Peter Ambros and Dr. Sabine Taschner-Mandl from the St. Anna Kinderkrebsforschung at the Children's Cancer Research Institute in Vienna, Austria. Austrian neuroblastoma patients were in part enrolled in the NB-A-87[57] or the HR-NBL-1/SIOPEN[58] trial (NCT01704716, EudraCT number 2006-001489-17). The copy number status of MYCN using FISH analysis was assessed as a routine diagnostic marker in all neuroblastoma clinical trials. Survival data for German patients of the discovery cohort was provided by Dr. Barbara Hero from the study office of the Neuroblastoma trial in Cologne. Additionally, a relapsed neuroblastoma cohort of patients enrolled in the registry trial INFORM[21] was analyzed (n = 40). Whole exome sequencing and low coverage whole genome sequencing was done as part of the INFORM workflow[21]. For a subset of these (n = 31) high coverage whole genome sequencing was done. A detailed overview of all clinical parameters and available data of the INFORM cohort is given in Supplementary Data 2. All patients or their legal guardian signed a written informed consent. Genomic DNA was isolated from fresh frozen material using a phenol chloroform extraction and control DNA was isolated from whole blood using the NucleoSpin Blood DNA extraction kit (Macherey-Nagel).

**C-Circle assay**. C-Circle assays were performed based on a previously described protocol[20], which is briefly described in the following. 30 ng of genomic DNA were used for every tumor sample. Prior to amplification, tumor DNA was digested with RsaI (4U/ug) and HinfI (4U/ug) and RNAse (25 ng/μg) treated for 1 h at 37 °C. 0.2 mg/mL BSA, 0.1% Tween 20, 1 mM dATP, dGTP and dTTP, 2 x Φ29 buffer and 7.5 U Φ29 polymerase was added to the pretreated DNA and rolling circle amplification was done at 30 °C for 8 h, followed by an inactivation at 65 °C for 20 min. For every sample, a sample without polymerase was included as control and on every blot the neuroblastoma cell lines CHLA-90 (ALT-positive) and SK-N-BE2 (ALT-negative) were included as positive and negative assay controls. TeloTTAGGG kit (Roche) was used for hybridization and development according to the manufacturers' instructions. All blots were quantified using the ImageJ gel quantification tool (ImageJ version 1.51). A C-Circle positive signal was defined as minimum 4-fold increase of the area under the curve relative to control without polymerase. In addition, the signal intensity had to be at least 20% of the CHLA-90 positive control. C-Circle status was assessed by quantification and optical inspection of all blots by two investigators.

**Whole genome sequencing**. High coverage whole genome sequencing libraries were sequenced on the Illumina HiSeq 2000 (100 bp paired end) or on a patterned flowcell v.2.5 (150 bp paired end) with coverage of 30–60x for the tumor and whole blood control samples. Tumor samples exhibited a tumor cell content of ≥60%. Sequencing libraries for the Illumina HiSeq 2000 platform were generated using the NEB Next Ultra DNA kit (NEB) according to the manufacturers' instructions. Libraries for sequencing on the patterned flowcell v2.5 were generated using the Truseq DNA Nano kit (Illumina) according to the manufacturers' instructions. Libraries were size selected using SPRI beads (Beckman Coulter Genomics). One Touch Pipeline (OTP) service of the German Cancer Research Center (DKFZ)[59] was used for alignment and variant calling. Whole genome sequencing data was aligned to the 1000 Genomes project phase 3 assembly with decoy and PhiX contigs reference genome using BWA-MEM version 0.7.8, 0.7.8-r2.08 or 0.7.15 with option -T 0. Sambamba version 0.6.5 was used for merging and duplication marking and bam files were filtered using Samtools version 0.1.19. The SNV calling workflow is available at [https://github.com/DKFZ-ODCF/SNVCallingWorkflow]. Copy number variations were assessed using ACEseq version 1.2.8[60], which was additionally used to assess the quality of the sequencing data. The ACEseq workflow is available at [https://github.com/DKFZ-ODCF/ACEseqWorkflow]. Genes were defined as amplified if the copy number was at least 8-fold higher than the total copy number of the tumor sample. Any copy number losses that were not homozygous deletions were defined as a deletion. Any loss of heterozygosity (LOH) that was not also a deletion (e.g., copy-neutral LOH) was defined as LOH. Small insertions and deletions were determined based on an OTP in-house workflow[61], which is available at [https://github.com/DKFZ-ODCF/IndelCallingWorkflow]. SOPHIA version 1.0.16 or 1.2.16 was used for detection of structural variants. SV calling workflow is available at [https://github.com/DKFZ-ODCF/SophiaWorkflow]. Structural variants affecting ATRX and TERT were determined using SOPHIA in combination with a manual inspection in the Integrated Genomics Viewer (IGV)[62,63]. TERT promoter mutations (C228T and C250T) were specifically analyzed using less stringent criteria for mutation calling (mutated base in at least two reads, mutation frequency ≥20%). Candidate genes associated with telomere maintenance and biology were identified using the TelNet database[22] [https://malone2.bioquant.uni-heidelberg.de/fmi/webd/TelNet]. Genes encoding ATRX and DAXX interaction partners were extracted from the BioGRID database [https://thebiogrid.org/].

**Telomere length analysis**. The telomere content was calculated using the software tool TelomereHunter[64] version 1.0.1. TelomereHunter v1.0.1 is available at [https://www.dkfz.de/en/applied-bioinformatics/telomerehunter/telomerehunter.html]. Telomeric reads containing six non-consecutive instances of the four most common telomeric repeat types (TTAGGG, TCAGGG, TGAGGG, and TTGGGG) were extracted. For the further analysis, only unmapped reads or reads with a very low alignment confidence (mapping quality lower than 8) were used. Telomere read count was normalized by all reads in the sample with a GC-content of 48–52% and furthermore telomere content of tumor samples was normalized to the matching blood control. For tumor samples without a matching control, the mean telomere content of all controls sequenced with the same protocol in the same sequencing center was used for normalization.

**Subgrouping based on telomere maintenance mechanism (TMM group)**. All tumors were categorized into four groups according to the genetic evidence of an activated telomere maintenance mechanism (ALT, MNA, TERT) or the absence of such a mechanism (OTHER). Tumors being either C-Circle positive or having a log2 telomere content relative to the control sample >1 were classified as ALT. Tumors with an amplified MYCN gene were considered to use telomerase activation by MYCN (MNA). The TERT group comprises tumors with a structural variation affecting TERT or TERT promoter mutation. Tumors with a relative log2 telomere content above one and amplified MYCN were included in the heterogeneous group HET ALT/MNA despite being C-Circle negative.

**Telomeric repeat loci**. To find telomeric repeat loci we searched for tumor-specific discordant paired-end reads, where one end was an extracted telomere read and the other end was non-telomeric and uniquely mapped to a chromosome (mapping quality >30). In 1 kb regions containing at least four discordant reads in the tumor sample and none in the matching control, exact junction sites were defined by at least three split reads. The split reads had to contain at least one TTAGGG repeat. Regions with discordant read pairs in at least 15 control samples were excluded. Finally, the junction sites were visualized using the Integrative Genomics Viewer (IGV)[62,63] to identify and remove remaining false positives. 1q42.2 hotspot region was manually inspected to add automatically excluded telomeric repeat loci (NBD158 and NBD137). The pipeline used to detect telomeric repeat loci is available via Github [https://github.com/linasieverling/TelomereRepeatLoci].

**RNA-sequencing**. Using the Nucleo Spin RNA kit (Macherey Nagel) RNA was isolated from fresh frozen tissue according to the manufacturers' instructions. mRNA was isolated using poly A selection with Dynabeads (Thermo Fisher). 100 bp paired sequencing was carried out using the Illumina Hiseq 2000 and Hiseq 4000. Alignment to the reference genome (1000 Genomes project phase 3 assembly with decoy and PhiX contigs) was done using STAR Version 2.5.2b[65] with the parameters–sjdbOverhang 200–runThreadN 8–outSAMtype BAM Unsorted SortedByCoordinate–limitBAMsortRAM 100000000000–outBAMsortingThreadN=1–outSAMstrandField intronMotif–outSAMunmapped Within KeepPairs–outFilterMultimapNmax 1–outFilterMismatchNmax 5–outFilterMismatchNoverLmax 0.3–twopassMode Basic–twopass1readsN −1–genomeLoad NoSharedMemory–chimSegmentMin 15–chimScoreMin 1–chimScoreJunctionNonGTAG 0–chimJunctionOverhangMin 15–chimSegmentReadGapMax 3–alignSJstitchMismatchNmax 5 −1 5 5–alignIntronMax 1100000–alignMatesGapMax 1100000–alignSJDBoverhangMin 3–alignIntronMin 20–clip3pAdapterSeq AGATCGGAAGAGCACACGTCTGAAC TCCAGTCA–readFilesCommand gunzip -c. Sambamba version 0.6.5[66] was used for merging and duplication marking. Bam files were filtered with SAM tools version 1.3.1[67]. Read counts per gene were obtained with featurecounts version 1.5.1[68] by counting reads over exon features using GENCODE 19 as a gene model. Both reads of a paired fragment were used for counting and only unique alignments were considered by setting the quality threshold to 255. The read counting was performed in an unstranded manner. The workflow used for RNA-sequencing is available at [https://github.com/DKFZ-ODCF/RNAseqWorkflow]. For the calculation of TPM expression values, genes on chromosomes X, Y, MT and rRNA and tRNA genes were omitted when calculating total library abundance calculations to prevent library size estimation biases. Differential expression analysis was performed using DESeq2 version 1.18.1[69] to find differences between samples with chr1q42.2-1qter deletion and those without a specific deletion in this region. chr1q42.2-1qter deletion was defined as reduced copy number compared to the upstream neighboring segment. Low-count genes with less than 10 reads across the entire cohort were excluded. Gender and sequencing location were included as possible confounding variables. Filtered raw count data was used for the default analysis including size factor estimation, dispersion estimation, negative binomial generalized linear model fitting, and Wald statistics. Log fold change shrinkage was applied. The results were filtered afterwards to include only genes from 230 Mb to the end of chromosome 1.

**TERRA expression**. To determine TERRA expression, RNA-sequencing reads containing telomere repeats were extracted with TelomereHunter[64] version 1.0.1 using a threshold of 14 telomere repeats per 100 bp read length and otherwise

default settings. All extracted telomere reads were considered as TERRA reads independently of the mapping position. To normalize for different library sizes, the number of TERRA reads was divided by the total read count and multiplied by 1 million. (Note that RNA-sequencing was done using poly-A selection).

**ATRX per exon normalized RNA expression**. The paired-end RNA-sequencing reads were aligned to human genome reference hg19 using STAR version 2.4.1 tool[65] with restriction to only unique alignments. Gene expression counts per exon were computed using corresponding DEXseq[70] version 1.24.3 based on adjusted GENCODE 19 annotation. The computed per exon counts from ATRX were extracted and normalized using RPKM measurement via exon length. Subset of formed exonic parts (EP 13, 16, 17, 23, 31, 37, 38) with no coverage variance across samples was excluded from the initial gene model annotation for the main heatmap generation. Genomic coordinates for the DEXseq ATRX exon model are given in Supplementary Table 4.

**Proteome analysis**. Tumor samples were lysed with 2% SDS, 50 mM ammonium bicarbonate buffer and complete mini EDTA-free Protease Inhibitor Cocktail. Samples were homogenized at room temperature 15 s for 7 cycles with a 5 s pause using FastPrep-24™ 5 G Homogenizer and heated to 95 °C for 5 min. Freeze–thaw cycles were repeated seven times. Benzonase was added to each sample and the lysate was clarified by centrifugation at 16 rcf for 30 min at 4 °C. Protein concentration was then measured using the Bio-Rad DC Protein assay. From each sample, 50 µg of protein were taken in duplicate. Proteins were reduced with 10 mM DTT for 30 min and alkylated with 55 mM iodoacetamide for 30 min. Wessel–Flügge precipitation was performed as previously described[71]. The retrieved protein pellet was resuspended in 6 M Urea, 2 M Thiourea in 10 mM HEPES (pH 8). Proteins were then digested with LysC for 2 h; samples were diluted in 50 mM ammonium bicarbonate buffer to reach less than 2 M urea before trypsin was added for overnight digestion. The peptide solution was acidified and fractionated by SCX on StageTips as described previously[72]. Samples were then separated by reversed phase HPLC on a 2000 mm monolithic column and analyzed online by high-resolution mass spectrometry on a Q Exactive plus mass instrument as described previously[73]. The resulting raw files were analyzed using MaxQuant software version 1.5.5.1 with LFQ quantification and 'match between runs' function[74]. Data was searched against a Human Uniprot database (2014-10; [https://www.uniprot.org/]) and common contaminants[74] at a false discovery rate (FDR) of 1% at both the peptide and protein level. The resulting text files were filtered to exclude reverse database hits, potential contaminants, and proteins only identified by site. Data was imputed by random draw from a normal distribution with default parameters: 0.3 width and 1.8 down shift. Proteins associated with telomere maintenance and biology were identified using the TelNet database[22] [https://malone2.bioquant.uni-heidelberg.de/fmi/webd/TelNet].

**ATRX/DAXX knockdown experiments and Western blot**. NBL-S neuroblastoma cells were cultured in RPMI media supplemented with 10% FBS, 100 U/ml penicillin, and 100 µg/ml streptomycin. siRNAs were transfected in 6-well format using Lipofectamine® RNAiMAX (Life Technologies) according to the manufacturer's instructions. NBL-S cells were transfected with siRNAs targeting DAXX or ATRX as control at a final concentration of 25 nM. siRNA sequences are given in Supplementary Table 2. NBL-S cells were harvested 96 h after transfection and cell pellets were lysed in M-PER lysis buffer (Thermo Fisher). 30 µg of protein lysate were run on NUPAGE 3–8% Tris-acetate gels (Thermo Fisher). Primary antibodies were incubated overnight at 4 °C. ATRX antibody was obtained from Sigma (HPA001906; rabbit polyclonal) and used at a dilution of 1:1000. DAXX antibody was obtained from Abcam (ab32140; rabbit monoclonal E94) and used at a dilution of 1:5000. Secondary anti-rabbit HRP (Dianova 111-035-144; goat polyclonal) coupled antibody was used at a dilution of 1:2000. Vinculin-HRP coupled antibody (clone 7F9, Santa Cruz sc73614) was used as loading control at a dilution of 1:1000. A list of all antibodies is provided in Supplementary Table 3. Experiments were done in three biological replicates. Western blot images were quantified using the gel analysis function of Image J (version 1.51). P values were calculated using a two-sided Student's t-test with the function t.test in R.

**Combined interphase FISH for MYCN and telomeres**
*NBD151*. Tumor touch imprints or 5 µm frozen sections of primary tumors mounted on microscopy glass slides were fixed with 4% Rotifix (Roth) for 10 min at 4 °C and FISH was performed using a telomere PNA-Cy3 FISH probe (Dako/Life Technologies) according to the manufacturer's instructions and a MYCN-digoxigenin (dig) probe (BAC clone RP11-355H10). Briefly, slides were pretreated with 0.05% pepsin in 0.01 N HCl for 1 min at 37 °C, followed by dehydration (70%, 96%, 100% EtOH series). Probes were applied, denaturation was carried out at 80 °C for 5 min and hybridization at room temperature for 3 h. After immersion in 1xPBS, slides were washed in standard saline citrate (SSC) containing 70% formamide at 42 °C for 15 min and twice with 0.1xSSC for 5 min. The MYCN-dig probe was detected using a sheep polyclonal anti-dig-FITC antibody (dilution 1:100; Roche 11207741910). Slides were washed in PNA Wash-buffer (Dako) and dehydrated in an EtOH series. Sections were counterstained and mounted with Vectashield containing DAPI (Vector laboratories) and covered with glass coverslips. Images

were acquired using a Zeiss Axioplan2 microscope and the ISIS software version 5.7.4 (Metasystems).

*Matching primary of NBI26*. Amplification status of MYCN was determined as described previously[75] with MYCN Vysis LSI N-MYC (2p24) SpectrumGreen/CEP2 SpectumOrange and Vysis LSI N-MYC Spectrum orange. Activation of ALT was determined by identification of ALT-associated promyelocytic leukemia (PML) nuclear bodies (APB) by immunofluorescence in combination with FISH using Alexa-488-(TTAGGG)n PNA probes as described previously[6]. Images were acquired using Leica DM 5550B fluorescent light microscope system at 63x resolution. Full z-stacks were taken at 0.5 µm and projected using Cytovision software (Version 7.6).

*NBI5*. Tumor touch imprints were fixed and labeled as previously described[76]. Telomere labeling was done using the Telomere PNA FISH Kit/Cy3 (Agilent). For MYCN FISH DNA from BAC clone RP11-355H10 was labeled with FITC-dUTP by nick translation. Fluorescent images were captured with a Leica DMRA2 microscope equipped with a Leica DFC360 FX camera and analyzed with the Leica CW 4000 FISH software.

**Telomerase activity**. Subset of the tumors was part of a previous publication[1]. Telomerase activity was measured using the TeloTAGGG Telomerase PCR ELI-SAPLUS Kit (Sigma Aldrich), which is a PCR-based telomeric repeat amplification protocol (TRAP) enzyme-linked immunosorbent assay (ELISA) kit, according to the manufacturer's instructions. Approximately 10 mg of tissue were used per tumor.

**FACS analysis of tumors**. FACS analysis of neuroblastoma tumors was done as part of a previously published study[77].

**ChIP-sequencing**. ChIP-sequencing was performed based on a combination of the previously published protocols[78,79], which is briefly summarized in the following. Tumor tissue was cut in 30 µm slices and fixed with 1% formaldehyde for 10 min followed by quenching with glycine for 5 min. Approximately 30 mg of tissue were used for each IP. Tumor tissue was lysed by adding 120 µL lysis buffer (50 mM Tris-HCl (pH 8.0), 10 mM EDTA, 1% SDS, protease inhibitor), homogenized using a pestle, incubated on ice for 5 min and sonicated for 5 cycles in a Diagenode Bioruptor Pico (30 s ON, 30 s OFF). After centrifugation (10 min, 12,000 rpm, 4 °C), the supernatant was transferred into a sonication tube (Diagenode). The pellet was resuspended in 30 µL lysis buffer and centrifuged again. 150 µL of mRIPA-I buffer without SDS (10 mM Tris-HCl (pH 8.0), 1 mM EDTA (pH 8.0), 500 mM NaCl, 0.1% Na-Deoxycholate, protease inhibitor) were added to the combined supernatants and samples were sonicated for 40 cycles (30 s ON, 30 s OFF). 220 µL mRIPA-I buffer without SDS were added and centrifuged (10 min, 12000 × g, 4 °C). Supernatant was kept and pellet resuspended in another 220 µL mRIPA-I buffer without SDS. Centrifugation was repeated and supernatants combined. 75 µL/IP magnetic protein G beads (Dynabeads Invitrogen) were washed twice with binding/blocking buffer (PBS with 0.5% BSA and 0.5% Tween-20). 3 µg of antibody were added together with 300 µL of binding/blocking buffer and incubated for at least 1 h at room temperature while rotating. H3K36me3 (ab9050; rabbit polyclonal), K3K9me3 (ab8898; rabbit polyclonal), and H3K27ac (ab4729; rabbit polyclonal) antibodies were obtained from Abcam. H3K27me3 antibody was purchased from Active Motif (39155; rabbit polyclonal). All antibodies are listed in Supplementary Table 3. The beads were washed again with blocking/binding buffer and sonicated chromatin was added and incubated overnight at 4 °C while rotating. Beads were washed once with ice-cold RIPA buffer (10 mM Tris-HCl (pH 8.0), 1 mM EDTA (pH 8.0), 140 mM NaCl, 0.1% SDS, 0.1% Na-Deoxycholate, protease inhibitor), five times with ice-cold RIPA buffer (no protease inhibitor), twice with ice-cold RIPA-500 (10 mM Tris-HCl (pH 8.0), 1 mM EDTA (pH 8.0), 500 mM NaCl, 0.1% SDS, 0.1% Na-Deoxycholate, 1% Triton X-100), twice with ice-cold LiCl wash buffer (10 mM Tris-HCl (pH 8.0), 1 mM EDTA (pH 8.0), 250 mM LiCl, 0.5% NP-40, 0.5% Na-Deoxycholate), and once with ice-cold 1xTE buffer (10 mM Tris-HCl (pH 8.0), 1 mM EDTA (pH 8.0)). Samples were eluted in 50 µL of direct elution buffer (10 mM Tris-HCl (pH 8.0), 5 mM EDTA (pH 8.0), 300 mM NaCl, 0.5% SDS). 5% of the sonicated material were used as input control and processed in parallel from this step onwards. Samples were RNAse A treated (0.2 µg/µL) for 30 min at 37 °C. 2.5 µL proteinase K (10 mg/mL) and 1 µL glycogen (20 mg/mL) were added and incubated for 2 h at 37 °C. Samples were incubated at 65 °C overnight to reverse cross-linking. Purification was done using Agencourt Ampure XP beads (Beckman Coulter Genomics) at a 2.3 ratio. Library preparation was done using the NEBNext End Repair Module (NEB), NEBNext dA-Tailing Module (NEB), and NEBNext Quick Ligation Module (NEB). High-Fidelity PCR Master Mix (NEB) was used for PCR amplification and adapter addition. Libraries were size selected using AgencourtAmpure XP beads (Beckman Coulter). Sequencing was done using Illumina HiSeq2000 50 bp single end sequencing.

**ChIP-sequencing data analysis**. ChIP-sequencing data was analyzed using a custom pipeline implemented in Snakemake. Using the Trimgalore tool version 0.4

[https://github.com/FelixKrueger/TrimGalore] reads were trimmed and subsequently data was aligned to the hg38 reference genome using Bowtie2 version 2.3. Unmapped reads were not excluded from the bam files during alignment. Bamfiles were processed using the deepTools2 suite[80] version 3.0. Using bamCompare input files were subtracted. Signal files (bigwig) were obtained using the SES method for normalization. Peaks were called using MACS2 version 2.1 [https://github.com/taoliu/MACS] for H3K27ac or SICER[81] version 1.1 for H3K9me3, H3K27me3, and H3K36me3. ChIP-sequencing workflow is available at [https://github.com/hdsu-bioquant/chipseq_telomeres/]. Telomeric reads containing at least 8 telomeric repeats per 100 bp read length were extracted in ChIP and input samples using TelomereHunter[64] with the option -rt 8 –rl. Number of telomeric reads was normalized to the total number of mapped reads. Reads containing the canonical SatII (ATTCCATTCGATTCCATTCG) and SatIII (ATTCCATTCCATTCCATTCCATTCCATTCC) sequence[35] were extracted using Telomere Hunter[64] version 1.0.1. Number of satellite reads was normalized to the total number of mapped reads. Log2 enrichment of normalized telomeric/satellite reads in ChIP was calculated relative to input files. Samples with a fraction of reads in peaks below one were excluded from the analysis.

$$Tel(log2)enrichment = log2\left(\frac{telomeric\ reads\ in\ IP/total\ mapped\ reads}{telomeric\ reads\ in\ input/total\ mapped\ reads}\right)$$

$$SatII(log2)enrichment = log2\left(\frac{SatII\ reads\ in\ IP/total\ mapped\ reads}{SatII\ reads\ in\ input/total\ mapped\ reads}\right)$$

$$SatIII(log2)enrichment = log2\left(\frac{SatIII\ reads\ in\ IP/total\ mapped\ reads}{SatIII\ reads\ in\ input/total\ mapped\ reads}\right)$$

**Statistics**. Differences between various neuroblastoma subgroups in terms of telomere content, telomeric repeat loci, enrichment of ChIP signals, TPM, and LFQ expression values were tested with two-sided Wilcoxon rank-sum tests using R (R Foundation for Statistical Computing, version 3.5.1 or 3.6.1). All boxplots were generated using the function geom_boxplot() in the R package ggplot2 (version 3.2.0 or 3.3.1). Thus, all boxplots indicate the median value. The lower and upper hinge of the boxplots correspond to the 25th and 75th percentiles. The upper/lower whisker spans from the hinge to the largest/smallest value (values expanding a distance of 1.5 x inter quartile range are not considered). Additionally, individual samples are shown as jitter points. Dotplot showing quantification of ATRX protein levels after DAXX knockdown based on western blot images was done using the function geom_dotplot() in the R package ggplot2 (version 3.3.1). Horizontal line is representing the median value. P values for Western blot quantification were calculated using a two-sided unpaired Student's t-test in R with the function t.test. P values for differential RNA expression analysis were calculated with Wald tests within DESeq2 version 1.18.1[69]. P values for differential protein expression analysis were calculated with two-sided Student's t-test statistics using Perseus software version 1.5.5.3[82]. Correlation analysis was performed using ggpubr (version 0.3.0) or with the function cor in R. Enrichment analysis was tested by hypergeometric test on gene symbols of significantly different proteins and mRNA determined for this analysis by SAM-modified[83] (S0 = 0.01) Welch t-test were performed using R. Multiple testing correction by Benjamini–Hochberg method was performed with the R function p.adjust. Survival curves were generated using Kaplan Meier analysis in the R package survminer (version 0.4.7). P values for survival analysis were calculated using log rank tests. The optimal cut point to assess the prognostic impact of the number of telomeric repeat loci and telomere content was determined using Maximally Selected Rank Statistics with the criteria that at least 20% of the cases have to be in each group using the R package survminer (version 0.4.7). Associations of C-Circle presence with clinical features like age, sex, stage, risk classification was calculated using Fisher tests. Enrichment of TP53 pathway mutations (TP53, CREBBP, ATM, ATR, CDKN2A, and MDM2) and PTPRD mutations was calculated using a Fisher test comparing the occurrence in ALT-positive compared to ALT-negative tumors. Enrichment of RAS pathway mutations (HRAS, KRAS, NRAS, BRAF, RAF1, NF1, CDK4, CCND1) was calculated using a Fisher test comparing their occurrence in ALT-positive tumors in the discovery cohort compared to ALT-positive tumors in the INFORM cohort. HET cases and relapse cases of the discovery cohort were excluded. A Fisher test was used to calculate the enrichment of proteins mentioned in the Telnet database in the differentially expressed proteins of ALT-positive tumors. Fisher tests were done using R (version 3.6.1).

**Reporting summary**. Further information on research design is available in the Nature Research Reporting Summary linked to this article.

## Data availability

Whole genome sequencing, whole exome sequencing, ChIP-sequencing, RNA-sequencing, and proteome data has been archived at the European Genome-phenome Archive [https://www.ebi.ac.uk/ega/] and can be found under accession number EGAS00001004349. Part of the whole genome and RNA-sequencing data was previously published[6] and is available under the accession number EGAS00001001308. The data is deposited under controlled access. Data access can be obtained by contacting Frank

Westermann for EGAS00001004349 or Matthias Fischer for EGAS00001001308. Access to data requires a data transfer agreement. mRNA and protein expression data are available as datasets in the R2 database and can be interactively analyzed. mRNA expression data can be found under the name "Tumor Neuroblastoma ALT - Westermann − 144 - tpm - gencode19" [https://hgserver1.amc.nl/cgi-bin/r2/main.cgi?table=ps_avgpres_2010fwr144_gencode19]. Protein data can be found under the name "Tumor Neuoblastoma ALT (Protein) – Westermann – 34 – LFQ - fw2010prot" [https://hgserver1.amc.nl/cgi-bin/r2/main.cgi?table=ps_avgpres_fw2010prot34_fw2010prot]. All functional SNVs, INDELs, SVs and copy number alterations (only amplification and homozygous deletions) for all tumors in the discovery cohort are provided as Supplementary Data 7–9. The remaining data are available within the Article, Supplementary Information or available from the authors upon request. Source data are provided with this paper.

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

## Acknowledgements

We thank all the patients and their families for consenting to the use of tumor material for this study. Additionally, we want to thank the Neuroblastoma Biobank and the INFORM incoming lab for the support of the study. We acknowledge the DKFZ Genomics and Proteomics sequencing core facility headed by Stefan Wolf for the excellent high-throughput sequencing service and the DKFZ Omics IT and Data Management Core Facility (ODCF) headed by Ivo Buchhalter for data management and data processing. Especially, we thank Ingrid Scholz from the DKFZ Omics IT and Data Management Core Facility (ODCF) for great support with the data upload. We thank

Carolin Meyer for providing bioinformatic tools and scripts for copy number analysis. We thank the lab of Irina Solovei for support with FISH analysis of cryosections. This project was supported by the Berlin Institute of Health within the collaborative research project TERMINATE-NB (CRG04) to MS. Furthermore, this project was supported by the German Ministry of Science and Education (BMBF) within the project SYSMED-NB (FW and MS), by the DKFZ-Heidelberg Center for Personalized Oncology (HIPO) within the HIPO2-K09R project (FW), by the Deutsche Krebshilfe within the project ENABLE (FW) and by the National Center for Tumor Disease within the project NCT 3.0-ENHANCE (FW). The project has been co-funded by the Vienna Science and Technology fund (WWTF) and the City of Vienna through the project LS18-111.

## Author contributions

Conception and design: S.A.H., L.S., L.F., F.W. Administrative support, provision of study materials, patients (and sequencing data): M.F., D.T.W.J., E.P., O.W., S.M.P., S.T.M., P.A., B.H. Conduct of the experiments, data analysis, and interpretation: S.A.H., L.S., M.Z., U.H.T., M.N.H., K.K., K.O., K.R., M.S., B.B., L.F., M.G., Y.G.P., K.O.H., E.M.W., L.S., C.R. Bioinformatics Support: L.S., M.Z., U.H.T., B.B., K.O., C.H., N.I., R.V., J.K., L.F. Manuscript writing: S.A.H., L.S., F.W. S.A.H. and L.S. contributed equally to the work. All authors read and approved the final manuscript.

## Funding

## Competing interests

The authors declare no competing interests.
