## [Peer Review File · Nature Communications]

REVIEWER COMMENTS

Reviewer #1 (Remarks to the Author): Neuroblastoma genomics

The ms. of Hartlieb et al. describes the analysis of two neuroblastoma series by DNA sequence, mRNA and proteomics analyses. The series are extensive and analysed in detail. This report focus on ALT-positive tumors. A number of interesting observations are reported. However, the paper is essentially descriptive and most of the interesting observations are not further investigated. Many observations are reported that have no clear implications. E.g. proteomics analysis showed reduced DAXX protein levels in ALT-positive ATRX wild-type tumors, suggesting that low DAXX protein levels lead to decreased ATRX protein levels and ALT. The interesting question would then be why DAXX protein levels are reduced in these tumors, but this question is not addressed. Overall, results of proteomic analyses are disappointing and most of this section is not informative.

Reviewer #2 (Remarks to the Author): ALT expert

This study provides a comprehensive and thorough analysis of cohorts of childhood neuroblastoma independently of the ALT status. The data from this study not only confirm many previous findings on ALT+ tumors, but also add new and important information. For example, they found that ATRX protein is generally reduced in ALT+ tumors independently of ATRX mutations. In addition, the epigenomic, proteomic, and sequencing analyses of this study reveal new features of ALT+ tumors. In particular, they found hotspots of intrachromosomal telomeric repeats in ALT+ tumors, which associate with poor clinical outcome of patients. The information generated by this study will be useful for further characterizing the mechanisms of ALT activation in tumors and the impacts of ALT on tumorigenesis. There are a few important questions about the data, interpretations and models of this study. This manuscript would be suitable for publication in Nat Commun if the questions below are satisfactorily addressed.

1. In Fig. 2c, it is surprising that some ALT tumors were negative for TERRA. Could these be false negatives for technical reasons?
2. In Fig. 3a, it is surprising that TERT expression is inversely correlated with telomere content. Many tumors with relatively high TERT expression have very low telomere contents, whereas some tumors with low TERT expression and no C-circle have intermediate levels of telomere contents. Can this be explained?
3. The interpretation of Fig. 4c is a little unclear. In ALT+ tumors with wild-type ATRX, why is DAXX protein lower without a reduction in DAXX RNA? Although the reduction of DAXX protein may explain the reduction of wild-type ATRX protein, why DAXX protein itself is reduced is not explained.
4. In Fig. 5b, is SUV39H1 expression altered in ALT+ tumors?
5. In Fig. 6, is microhomology to telomeric repeats found at 18q23 and 19q13.43?
6. All three hotspots of intrachromosomal telomere loci are very close to chromosome ends. What about the intrachromosomal telomeric repeats far from chromosome ends? Do their breakpoints also have microhomology to telomeric repeats? Some of the less frequent telomeric repeat loci are very close to chromosome ends too (on Chr1, some loci are even closer to telomeres than 1q42.2). Do the breakpoints of these loci have less homology to telomeric repeats? What determines the frequency of neo-telomere formation?
7. For the intrachromosomal telomeric repeats far from chromosome ends, are they flanked by non-telomeric sequences on "both sides"? How would the "neo-telomere" model explain this type of loci? This type of loci seems to increase in ALT_ATRXwt tumors as well. Are they generated through a mechanism different from those hotspots?

8. Fig. 4 suggests that ATRX protein could be down regulated independently of mutations. In Fig. 6, it would be helpful to compare the levels of telomeric repeat loci between ATRX-high and ATRX-low tumors. Loss of ATRX protein may be a better marker for intrachromosomal telomeric repeats.

Reviewer #3 (Remarks to the Author): ALT expert

The paper entitled "Alternative lengthening of telomeres in childhood neuroblastoma from genome to proteome" by Hartlieb et al. aimed at developing studies that could potentially further document the ALT (Alternative Lengthening of Telomeres) pathway (an alternative mechanism to telomerase reactivation, based on telomeric recombination, that allows tumor cells to maintain functional telomeres) in neuroblastoma, the most common extracranial solid tumor occurring mostly in early childhood.

In neuroblastomas, both the reactivation of telomerase pathway and activation of the ALT pathway are synonymous for dismal outcome, as observed in previous studies, and this occurs in around half the cases. On the other hand, the other half of cases show no sign of reactivation of telomere maintenance (either telomerase or ALT) and these tumors have an excellent outcome, either spontaneously regressing or differentiating into benign ganglioneuromas. Here, the authors focused their studies on further documenting neuroblastoma ALT by analyzing various biological and genetic features from large collections of tumors, always comparing with MYCN-amplified tumors and telomerase-sustained tumors.

The work produced by these authors is really outstanding and exemplary. The authors' approach is really impressive, because it allowed to produce very large amounts of data using RNA sequencing, whole genome sequencing, ChIP-seq, mass spectrometry for whole proteome analysis, analyses of telomere sequences content, telomere length, ALT activity by C-circle assay, and analyses of epigenetics pathways.

An important question associated with neuroblastoma is to understand why tumors with no sign of telomere maintenance mechanism are more benign than tumors with either telomerase- or ALT-maintained tumors, although it may be obvious that these tumor cells eventually lack the potential to maintain chromosomes stable enough to allow continuous proliferation. On the other hand, it is also important to understand the biological features of ALT tumors compared with those of MYCN- and telomerase-tumors in order to better dictate appropriate therapy. The present study provides us with a pretty good number of new findings concerning neuroblastoma ALT characteristics without however making any major breakthrough in this field. These characteristics may or may not apply to other types of ALT cancers. Nevertheless, no doubt that these new findings will be of a great help to better understand neuroblastoma, but also all the other types of ALT cancers.

I have a number of remarks and questions, exposed below.

1-The percentage of ALT tumors known to be around 10% in previous studies, and in the present one as well (in the cohort of 718 children), was found here to increase to 47.5% when a particular collection of neuroblastoma tumors, namely one composed of relapsed cases, was screened for ALT. I understand that ALT positivity was associated with older age at diagnosis in these relapsed cases as well as in the screening cohort (lines 72-73). From these 47.5% vs 10%, one can understand that either ALT becomes more frequent as the tumors evolve with time, or, alternatively, that relapse triggers some sort of pathway favoring ALT activation. Do the authors have additional information that could potentially allow to favor one hypothesis or the other? In addition, the authors observed that "relapsed ALT tumors showed increased telomere content compared to the matching primary sample" (lines 122-123). Therefore, ALT products and/or consequences appear to accumulate with age. On the other hand, there is a large number of analyzed relapsed tumors that evolved from MYCN or telomerase tumors (hence the increase of ALT tumors from 10 to 47.5%). C-circle measurement can also provide the intensity of the ALT pathway, as you know. Therefore, it would be very informative to provide the average value for C-circle intensity in the relapsed tumors and compare with that in the 66 CC+ tumors of the screening cohort to see if only those previously ALT+ tumors relapsed to ALT+ have increased ALT activity (either increased telomere content or C-circle value) or if also telomerase tumors relapsed to ALT+ tumors also have high ALT activity.

2- Lines 210-213: Concerning the enrichment of ALT activity in relapsed tumors, the authors suggest that the unfavorable outcome of these tumors might be due to a “primary resistance to the current standard treatment regimens targeting strongly proliferating tumors, which are probably not suited to treat slowly growing ALT tumors, particularly those with canonical activating RAS pathway mutations”.

It would be important to analyze the treatment received case by case by the children of the INFORM cohort. Indeed, if an inappropriate treatment, namely against strongly proliferating tumors, was applied to ALT tumors, it will show up in these numbers, because it is known whether the initial status of the relapsed tumors was ALT, or telomerase, or MYCN or no TMM. In fact, I do not understand how treatment may have been ill appropriate at first diagnosis, because, initially, ALT tumors were recognized as such and were probably not treated with drugs targeting strongly proliferating tumors, unless inappropriateness of the treatment was not recognized at the moment. If inappropriateness of the treatment was known, then it was not likely applied and, therefore, the unfavorable outcome of ALT tumors cannot be due to non appropriate treatment.

3- Lines 76-77: Do you know why the “Amplified MYCN and C-Circle presence were mutually exclusive in the screening/discovery cohort, but not in the INFORM cohort (Fig. 1b)”? Since the MYCN amplified C-circle positive INFORM tumors concern only two patients (out of the 19/40 positive ones), the significance of this absence of mutual exclusion between MYCN amp and C-circle + may not be really due to a well defined molecular event. Have the authors thought of a possible explanation for this? If MYCN-amp and ALT are really present together in these two tumors, can these identify a particular subtype of neuroblastoma in which ALT and telomerase activation could co-exist?

4- Lines 152-154: “Interestingly, ALT-positive ATRX wild-type tumors exhibited reduced protein levels of the ATRX interaction partner DAXX (Fig. 4c), pointing towards an alternative route of loss of ATRX/DAXX complex activity”. This was observed in all 5 ATRX-wt ALT+ tumors tested. Do you think that this observation can now allow to conclude that there is no other type of ALT tumor (in neuroblastoma at least) besides the ATRX-mutated tumors (55%) and the ATRX-wt DAXX low expressed tumors and that ALT always results from ATRX-DAXX dysfunction? If yes, this would be very informative to other researchers willing to check this in other types of ALT cancers.

5- Lines 92-93 and Suppl. Fig. 2: Have the authors examined in even much more detail the presence of mutations in the ALT+-ATRX-mut versus ALT+-ATRX-wt tumors and, more precisely, analyzed possible correlations between the two groups of tumors? I'll explain. As I understand, the around 45% of ALT-ATRX-wt tumors of the discovery cohort suffer from very affected DAXX levels (5/5, Fig. 4c). Therefore, one might think, as I said in my precedent point and as you suggest, that in these ALT+-ATRX-wt tumors, the reason for the presence of ALT activity is also due to the absence of a functional ATRX-DAXX complex and, therefore, all types of ALT tumors would have the same origin, a loss of functional ATRX-DAXX complex activity. My question is: If the reason for the existence of ALT activity is only due to loss of either ATRX or DAXX, we should expect all other mutations, shown in Suppl. Fig. 2, to be rather similar in the two groups, which is not the case. How can we explain that? And, in addition, have you looked whether these differences in mutations between the two groups could give clues to understand the origin of the decrease in DAXX levels in these tumors? Or, else, to understand the type of post-transcriptional regulation of ATRX you suggest (based on actual observations)?

On the other hand, these 5 cases of decreased DAXX levels might not be numerous enough to extrapolate and conclude that all ALT+-ALT-WT tumors are ALT positive because of reduced DAXX levels? Or, else, the extent of decrease might not be large enough to provoke total loss of function of the ATRX-DAXX complex?

6- Lines 100-101: You stated that overall survival was not significantly different between ATRX-mutated and ATRX wild-type ALT cases (Fig. 3c-d). Yet, in Fig. 3c, the survival of the ALT+-ATRX-wt patients after ~ 12 years seems to be much more than that in the ALT+-ATRX-mut patients. None of the latter group survived after ~ 14 years, unlike in the former group.

7- It was surprising to learn that POLD3 was mutated in a substantial number (50%) of ALT tumors (Suppl. Fig. 4), as POLD3 has recently been reported to be essential for ALT in human

tumors (Dilley et al, Nature, 539 (2016) 54-58; Roumelioti et al, EMBO Rep, 17 (2016) 1731-1737). Had the authors noticed this point? And would they like to comment on it? In addition, this POLD3 mutation is substantiated by the fact that you also found by mass spec that POLD1, another subunit of DNA polymerase delta had diminished levels (entry #182 of Suppl. Table 4), as had POLDIP3, a POLD3-interacting protein (entry #253).

8- There is an apparent inverse correlation between the frequency of telomeric repeat loci and the protein level of EXO1. Indeed, the telomeric repeat frequency was significantly higher in the ALT+-ATRX-mut tumors than in the ALT+-ATRX-WT tumors (Fig. 6a), while, on the other hand, EXO1 protein levels were lower in the ALT+-ATRX-mut tumors than in the ALT+-ATRX-WT tumors (Suppl. Fig. 10). Could this correlation be real? And if yes (but how to verify this?), how could it be explained?

Perhaps I did not completely understand the mechanistic of telomere repeat insertion, but in my mind, loss of a chromosome fragment (such as that containing EXO1 at 1q42.2) will always correspond to one telomere repeat Insertion event and, therefore, the increased frequency of telomere repeat in ALT+-ATRX-mut tumors cannot in itself explain the diminution of EXO1 expression in these tumors.

Reviewer #4 (Remarks to the Author): Proteomics expert

Hartlieb and colleagues use multi-omics approaches to comprehend the molecular features characterizing neuroblastomas, in particular ALT positive tumors from large patient cohorts. Unlike telomerase, it is difficult to 'quantify' ALT activity, if that is a biologically relevant thing to do. Here the authors score ALT in neuroblastomas by detecting C-Circles, one of the most reliable, but not definitive, ALT marker. They perform this on a screening cohort (n=718 tumors) and on the INFORM cohort (n=40). They performed whole genome sequencing, transcriptomics, ChIP sequencing and whole proteome analysis, and looked for correlations between ALT (C-circle presence) and their omics dataset and clinical data. From these data they confirm that ALT neuroblastomas are biologically distinct from telomerase positive tumors. The main observed characteristics from the ALT tumors are the following:

- (i) higher telomeric DNA content than in ALT negative tumors
- (ii) low TERT RNA expression, TERRA accumulation
- (iii) a complex mutational landscape. A rather high ATRX mutation rate, DAXX and H3F3A mutations are rare. Importantly, ATRX wt tumors can display ALT features, however, it is unclear if ATRX function is 'normal' in these cases: e.g. some cases show normal ATRX mRNA expression but low ATRX protein levels.
- (iv) telomeric H3K9me3 enrichment and H3K27me3 depletion
- (v) high frequency of ALT specific interstitial telomeric repeats which are unstable, and identification of recurrent telomeric amplified repeat hotspots, for example at the 1q42.2 chromosomal position.

The study is of high quality and very interesting because of the size of the screening cohort and the inclusion of relapsed tumors (INFORM). The study is essentially confirmatory and descriptive for the most parts. Nonetheless, I think it will be useful for researchers in the fields of cancer and telomere biology. Therefore, I am supporting the publication of this article, pending many minor revisions.

- (1-) Raw C-Circle blots are missing. Would it be possible to show these, at least for a subset of representative tumors ?
- (2) Telomerase assays, which are quite straightforward, should be attempted also on a subset of representative tumors
- (3) Previous published data have shown that ALT is not present in MYCN-amplified tumors, and that MYCN amplification and ATRX mutation are generally exclusive. How do the authors explain the co-occurrence of MYCN amplification with some ALT markers (detected C-Circles and a slight increase of telomere content but ATRXwt) in the INFORM cohort (NBI5-NBI26)? Could it be explained by the acquisition of one or the other TMM at the relapse? Can it be ALT independent? For instance, replication stress and trimming dysfunction can generate extrachromosomal Circle accumulation independently of the ALT pathway.
- (5) "reduced ATRX-DAXX complex activity" (l.47), "loss of ATRX/DAXX complex activity" (l.153),

"reduced protein complex activity" (l.205): no experiment has been performed to measure activity or the genome wide binding of these proteins. The authors only observe decreased protein levels by LFQ proteomics, they should avoid these inappropriate statements.

(6) The parameters used for ChIPseq telomeric sequences searching by TelomereHunter are not available in the material and methods.

(7) H3K9me3 enrichment, H3K27me3/Ac depletions at ALT telomeres. Sometimes the log2 enrichment values are below zero, which indicate very weak signals. But the problem with this representation is that one cannot estimate if the telomeres are truly enriched or depleted of these marks compared to other known loci (unique or repeated) where these marks are known to be present: like pericentromeric regions for H3K9me3 (alpha-satellites), or the telomerase promoter for H3K27me3 (values that you have reported in another figure). H3K27ac and H3K27me3 enrichments at satellite alpha and III are also a good negative control and give an idea of the background than histone ChIP sometimes have. The authors should add these analyses.

(8) (l.125-128) Unclear: In Sup. Fig.6a the authors present data from the Inform cohort (NBI) and not from the discovery cohort (NBD). However, in the sentence they compare NBI and NBD for RAS mutation. Moreover, the word "supporting a specific impact of aberrant RAS signaling in relapsed ALT positive tumors" is not appropriate, no biological experiments have been done to confirm dysfunctional RAS signaling in those tumors.

(9) "many proliferation-associated proteins" (l.143), the authors should be more precise, how many? Which pathways? not clear in the Sup. Fig.7a.

(10) The authors conclude "ALT tumors exhibited a high rate of neo-telomere formation" (l.207) There is no experimental data showing this conclusion. They suggest this hypothesis before in the text, but the loss of chromosomal copy number correlated with telomeric insertion could be also the result of chromosome internal ITS instability, these interstitial repeats may promote fragile hotspots prone to breakage in ALT cells.

Point-by-point response

We thank all the reviewers for their time and favorable consideration. We highly appreciate the constructive suggestions and comments, which substantially improved our manuscript. Our response to the individual comments is outlined below.

Reviewer #1 (Remarks to the Author): Neuroblastoma genomics

The ms. of Hartlieb et al. describes the analysis of two neuroblastoma series by DNA sequence, mRNA and proteomics analyses. The series are extensive and analysed in detail. This report focus on ALT-positive tumors. A number of interesting observations are reported. However, the paper is essentially descriptive and most of the interesting observations are not further investigated. Many observations are reported that have no clear implications. E.g. proteomics analysis showed reduced DAXX protein levels in ALT-positive ATRX wild-type tumors, suggesting that low DAXX protein levels lead to decreased ATRX protein levels and ALT. The interesting question would then be why DAXX protein levels are reduced in these tumors, but this question is not addressed. Overall, results of proteomic analyses are disappointing and most of this section is not informative.

We thank the reviewer for his time and comments and are glad the reviewer finds that our analysis is “extensive” and “interesting”. We agree that our manuscript is mainly descriptive in the sense that it describes genomic, transcriptomic and proteomic features of ALT-positive neuroblastomas. However, it comprises to our knowledge the largest cohort of ALT-positive neuroblastoma tumor samples, including a cohort of relapsed neuroblastomas from the clinical sequencing registry trial INFORM. Further, we analyzed and described the dataset in exceptional detail using various omics layers. Using the presented analysis, we could provide evidence that ALT-positive neuroblastomas form a separate subgroup of high-risk tumors clinically as well as biologically. In the current clinical practice, these tumors are treated with the same therapy approach as telomerase-activated neuroblastomas. We believe that our work contributes to a better understanding of the molecular and clinical features of ALT-positive neuroblastomas. The increased insight into this subgroup is necessary to develop urgently needed revised treatment strategies for these children in the next step.

We apologize for not describing the proteomics part in enough detail, but disagree with the statement that the proteomic analysis is “not informative”. We provide the first neuroblastoma proteome analysis. Concerning technical aspects, our proteomics analysis is state of the art and of high quality. Importantly, we only observe reduced ATRX/DAXX levels only at the protein but not at the mRNA level. Hence, the proteomic data provides evidence for the ALT phenotype while the transcriptomic data does not, which means that the proteomic data is in fact more informative than the transcriptomic data in this regard.

We fully agree with the reviewer that it would be very interesting to identify the exact mechanisms behind the reduced ATRX/DAXX protein levels. Our data suggests that these mechanisms differ between tumors. First, *ATRX*-mutated and *ATRX* wild-type tumors appear to behave differently, as can be seen for the reduced DAXX protein levels in the latter group. Second, there does not seem to be a single mutation common to all ALT-positive *ATRX* wild-type tumors, indicating that different mechanisms can lead to reduced ATRX/DAXX protein levels also in this subset. While we cannot pinpoint the exact molecular mechanism(s) involved, we do provide evidence that the reduced ATRX levels in ALT-positive *ATRX* wild-type tumors could result from the degradation of orphan ATRX proteins, which could be the result of reduced DAXX levels.

We have further significantly extended the section on the proteomics analysis in the results and discussion part. We feel that these additions clarify the points raised by the reviewer and highlight key questions for future research. We are confident that the revised version of our manuscript has

been improved regarding many points of concern and hope to convince reviewer #1 that it is suitable for publication in Nature Communications.

The revised results text on the proteomic analysis now reads (note that reference numbers were adapted to match the order of references in the point-by-point response and differ from the original text):

“Using a mass spectrometry based whole proteome approach (n = 34) quantifying on average 6,891 proteins per tumor, we found 470 proteins significantly different in ALT-positive tumors compared to the other subgroups (nominal P ≤ 0.01; fold change ≥ 2; Fig. 4a, Supplementary Data 5). Enrichment analysis of significantly different mRNAs and proteins for each subgroup in turn highlighted both the overall agreement of transcriptomics and proteomics as well as the distinct effects only significantly present in one or the other omics-level (Fig. 4b, Supplementary Fig. 11a, Supplementary Data 6). The Ras family proteins HRAS and NRAS, comprised in the term “prenylation”, were significantly upregulated in ALT-positive neuroblastomas (Fig. 4b, Supplementary Data 6.). Among the top upregulated proteins in ALT-positive tumors were the cancer testis antigens P antigen family member 5 (PAGE5) and melanoma associated antigen 4 (MAGEA4), which may serve as potential targets for immunotherapies (Fig. 4a). Many pathways associated with proliferation were significantly downregulated on protein level in ALT-positive tumors compared to the telomerase-activated tumors including “DNA replication” and “chromosome”. Notably, the classical proliferation marker MKI67¹, comprised in the term “chromosome”, was one of the top significantly downregulated proteins in ALT-positive neuroblastomas (Fig. 4a-b, Supplementary Fig. 11b). The term DNA replication includes among others the proteins PCNA, MCM2, MCM3, MCM4, MCM5 and MCM7, which are also indicative of the proliferative capacity of a cell². Furthermore, the term “bromodomain” was significantly reduced in ALT-positive tumors, while strongly upregulated in MNA tumors. Amplified MYCN is known to be associated with increased occupancy of active promoter regions and enhancer invasion by MYCN and bromodomain proteins, leading to an increased transcriptional activity of many proliferation associated genes and downregulation of differentiation genes³. Moreover, ALT-positive tumors exhibit a significantly lower fraction of cycling cells compared to MYCN-amplified tumors (Supplementary Fig. 11c). Taken together, both protein expression data and cell cycle analysis support a low proliferative capacity of ALT-positive neuroblastomas. 96 of the 470 significantly different proteins were annotated in the TelNet database to be associated with telomere maintenance⁴ (Supplementary Data 5) (P = 0.03). The significantly downregulated proteins in ALT-positive neuroblastomas comprised ATRX, PCNA, EXO1, GNL3, KDM1A, RIF1, SCG2, TFAP2B and GNL3L, all functionally linked to telomere maintenance⁴. Helicases and chromatin regulators including ATRX, were less abundant at protein level in the ALT subgroup, but not at mRNA level (Fig. 4b). Importantly, ATRX itself was among the top candidates to exhibit lower protein abundance in ALT-positive tumors (Fig. 4a). Intriguingly, the reduced ATRX protein levels were independent of ATRX mRNA levels and mutation status (Fig. 5a). ATRX was among the most strongly down-regulated proteins while the ATRX mRNA changes very little (Supplementary Fig. 11d), indicating that reduced ATRX protein abundance is a characteristic proteomic feature of all ALT-positive tumors (ATRX wild-type and mutated) that cannot be observed at the mRNA level. Analysis of exon-specific ATRX mRNA levels revealed that only the exons affected by deletion have reduced expression in the ATRX-deleted cases (Supplementary Fig. 12a). In summary, reduced ATRX protein level is a biomarker for ALT-positive neuroblastomas that is independent of mRNA levels and ATRX mutation. The decreased ATRX protein levels despite unchanged mRNA abundance could be due to reduced translation and/or increased degradation of ATRX in ALT-positive tumors. For example, it has been shown that unassembled subunits of multiprotein complexes (so-called orphans) are often degraded⁵. Downregulating one subunit can therefore induce degradation of other subunits, which partially explains divergence of protein- and mRNA level changes in many biological contexts^{6,7}. We

therefore took a closer look at the ATRX binding partner DAXX. Interestingly, DAXX protein levels were specifically reduced in ALT-positive ATRX wild-type tumors (Fig. 5b). To investigate if reduced DAXX levels could explain ATRX down-regulation at the protein level, we knocked-down DAXX in the ALT-negative neuroblastoma cell line NBL-S and observed that ATRX protein levels were indeed reduced after 96 h (Fig. 5c-d). Hence, the reduced ATRX protein levels in ATRX wild-type ALT positive tumors might result from down-regulation of the DAXX protein. However, since DAXX mRNA levels do not differ significantly between tumors (Fig. 5b) and no recurrent mutation patterns in DAXX or ATRX/DAXX interacting proteins could be identified in ALT-positive ATRX wild-type tumors (Supplementary Fig. 12b, 13), the question what causes the downregulation of DAXX is still open. Hence, the mechanistic details behind ATRX/DAXX complex reduction remain to be uncovered.” (lines 170 – 225)

The revised discussion text on the proteomic analysis now reads (note that reference numbers were adapted to match the order of references in the point-by-point response and differ from the original text):

”Integrating proteomic profiling revealed reduced ATRX/DAXX protein complex abundance as recurrent event in ALT-positive neuroblastoma, which could often not be explained by mutations in these genes. The observation that all five ALT-positive tumors with wild-type ATRX depicted reduced DAXX protein levels is intriguing. It is tempting to speculate that the ALT phenotype always results from loss of the ATRX/DAXX complex activity, which is either caused by ATRX mutations or by reduced ATRX/DAXX protein levels. Future studies will show if this is indeed the case in neuroblastoma and/or possibly other ALT-positive cancers. Nevertheless, we could identify a subgroup of ALT-positive neuroblastomas with wild-type ATRX, but low ATRX protein abundance. Reduced ATRX protein levels in ALT ATRX wild-type neuroblastoma could be explained by the reduced DAXX protein levels, which we specifically observed in this subgroup of tumors. In this scenario, reduced DAXX protein levels impair assembly of the ATRX/DAXX complex, which then causes degradation of orphan ATRX protein molecules. Consistent with this idea, we observed that knocking down DAXX reduced ATRX protein levels in neuroblastoma cells. However, the cause of reduced DAXX protein levels is not yet clear, especially since mRNA levels do not change significantly. Reduced DAXX protein levels in ALT-positive tumors may result from posttranscriptional events. It is known that DAXX is regulated via various posttranslational modifications including phosphorylation, SUMOylation and ubiquitination⁸. Irrespective of the mechanistic details involved, our study highlights that proteomic data is closer to phenotypes than transcriptomic data, which is especially valuable in a clinical context⁷.” (lines 304 – 322)

Reviewer #2 (Remarks to the Author): ALT expert

This study provides a comprehensive and thorough analysis of cohorts of childhood neuroblastoma independently of the ALT status. The data from this study not only confirm many previous findings on ALT+ tumors, but also add new and important information. For example, they found that ATRX protein is generally reduced in ALT+ tumors independently of ATRX mutations. In addition, the epigenomic, proteomic, and sequencing analyses of this study reveal new features of ALT+ tumors. In particular, they found hotspots of intrachromosomal telomeric repeats in ALT+ tumors, which associate with poor clinical outcome of patients. The information generated by this study will be useful for further characterizing the mechanisms of ALT activation in tumors and the impacts of ALT on tumorigenesis. There are a few important questions about the data, interpretations and models of this study. This manuscript would be suitable for publication in Nat Commun if the questions below are satisfactorily addressed.

We highly appreciate the positive feedback and the helpful comments.

1. In Fig. 2c, it is surprising that some ALT tumors were negative for TERRA. Could these be false negatives for technical reasons?

To assess a possible influence of sequencing quality, we analyzed the relation of TERRA read counts to the estimated library complexity for each sample (Supplementary Fig. 3c). The estimated library complexity of samples with at least four detected TERRA reads ranged between 8,674,013 read counts and 438,842,963 read counts. All four ALT-positive samples with zero TERRA reads had a library complexity within this range. However, one of the samples was within the lower edge of the range with a library complexity of 16,170,875 read counts. In general, the TERRA reads were analyzed using polyA selected RNA libraries. Since only about 10% of the TERRA molecules have a polyA tail⁹, we can only conclude on a fraction of the total TERRA levels. Therefore, the total number of TERRA reads per sample is rather low and it might be difficult to distinguish between a very low TERRA expression and zero TERRA expression.

Supplementary Figure 3c:

Number of raw TERRA read counts relative to the estimated library complexity. Color coding indicates TMM group.

2. In Fig. 3a, it is surprising that TERT expression is inversely correlated with telomere content. Many tumors with relatively high TERT expression have very low telomere contents, whereas

some tumors with low TERT expression and no C-circle have intermediate levels of telomere contents. Can this be explained?

Indeed, *TERT* expression is negatively correlated with telomere content (Spearman correlation $r = -0.56$, $P = 2.75e-13$). ALT-positive neuroblastomas have low *TERT* expression and a high log2 telomere content tumor/control ratio, meaning that the telomere content of the tumor is higher than the telomere content of the normal/blood control. *TERT*-activated neuroblastomas, being activated due to either amplified *MYCN* or rearranged *TERT*, have a lower telomere content in the tumor compared to the control samples resulting in a negative log2 telomere content tumor/control ratio. Telomere maintenance via *TERT* activation is associated with fast tumor progression and highly aggressive tumor growth¹⁰, thus these tumor cells proliferate very fast. Fast proliferation most likely results in rather short telomeres, since telomerase does not have much time to extend the telomeres before the next cell division. Therefore, we assume that *TERT*-activated neuroblastomas have short telomeres that are constantly extended and kept at the low, but sufficient length. Tumors showing no evidence of an activated telomere maintenance mechanism (OTHER group) have roughly the same telomere content in the tumor and in the control and have minimum or low *TERT* expression. Tumors without a telomere maintenance mechanism have a good outcome¹⁰ due to very slow tumor progression and limited growth. Our results are also consistent with previous publications on the telomere length of *TERT*-activated tumors^{10,11}.

Additional Figure 1: *TERT* RNA expression relative to telomere content.
Linear regression line in grey. Spearman correlation $r = -0.56$, $P = 2.75e-13$.

3. The interpretation of Fig. 4c is a little unclear. In ALT+ tumors with wild-type ATRX, why is DAXX protein lower without a reduction in DAXX RNA? Although the reduction of DAXX protein may explain the reduction of wild-type ATRX protein, why DAXX protein itself is reduced is not explained.

We apologize for not making this clearer. The reviewer is right in pointing out that we cannot provide a mechanistic explanation for the reduction in DAXX protein levels in the *ATRX* wild-type tumors. We now made these points clearer in the results and discussion section. We propose that DAXX expression is regulated on the posttranscriptional level. DAXX mRNA was not altered in ALT-positive tumors and we could also not identify any recurrent mutations that affect DAXX directly or indirectly (Supplementary Fig. 12b, 13). DAXX expression might be regulated by post-

translational modifications or ubiquitin-mediated degradation, since DAXX can be modified by various post-translational modifications⁸. However, the experimental proof of such modifications in our neuroblastoma tumor tissues is not possible due to the limitation of tumor material and is beyond the scope of this manuscript.

From the results section (note that reference numbers were adapted to match the order of references in the point-by-point response and differ from the original text):

"Importantly, ATRX itself was among the top candidates to exhibit lower protein abundance in ALT-positive tumors (Fig. 4a). Intriguingly, the reduced ATRX protein levels were independent of ATRX mRNA levels and mutation status (Fig. 5a). ATRX was among the most strongly down-regulated proteins while the ATRX mRNA changes very little (Supplementary Fig. 11d), indicating that reduced ATRX protein abundance is a characteristic proteomic feature of all ALT-positive tumors (ATRX wild-type and mutated) that cannot be observed at the mRNA level. Analysis of exon-specific ATRX mRNA levels revealed that only the exons affected by deletion have reduced expression in the ATRX-deleted cases (Supplementary Fig. 12a). In summary, reduced ATRX protein level is a biomarker for ALT-positive neuroblastomas that is independent of mRNA levels and ATRX mutation. The decreased ATRX protein levels despite unchanged mRNA abundance could be due to reduced translation and/or increased degradation of ATRX in ALT-positive tumors. For example, it has been shown that unassembled subunits of multiprotein complexes (so-called orphans) are often degraded⁵. Downregulating one subunit can therefore induce degradation of other subunits, which partially explains divergence of protein- and mRNA level changes in many biological contexts^{6,7}. We therefore took a closer look at the ATRX binding partner DAXX. Interestingly, DAXX protein levels were specifically reduced in ALT-positive ATRX wild-type tumors (Fig. 5b). To investigate if reduced DAXX levels could explain ATRX down-regulation at the protein level, we knocked-down DAXX in the ALT-negative neuroblastoma cell line NBL-S and observed that ATRX protein levels were indeed reduced after 96 h (Fig. 5c-d). Hence, the reduced ATRX protein levels in ATRX wild-type ALT positive tumors might result from down-regulation of the DAXX protein. However, since DAXX mRNA levels do not differ significantly between tumors (Fig. 5b) and no recurrent mutation patterns in DAXX or ATRX/DAXX interacting proteins could be identified in ALT-positive ATRX wild-type tumors (Supplementary Fig. 12b, 13), the question what causes the downregulation of DAXX is still open. Hence, the mechanistic details behind ATRX/DAXX complex reduction remain to be uncovered." (lines 200 – 225)

From the discussion section (note that reference numbers were adapted to match the order of references in the point-by-point response and differ from the original text):

"Integrating proteomic profiling revealed reduced ATRX/DAXX protein complex abundance as recurrent event in ALT-positive neuroblastoma, which could often not be explained by mutations in these genes. The observation that all five ALT-positive tumors with wild-type ATRX depicted reduced DAXX protein levels is intriguing. It is tempting to speculate that the ALT phenotype always results from loss of the ATRX/DAXX complex activity, which is either caused by ATRX mutations or by reduced ATRX/DAXX protein levels. Future studies will show if this is indeed the case in neuroblastoma and/or possibly other ALT-positive cancers. Nevertheless, we could identify a subgroup of ALT-positive neuroblastomas with wild-type ATRX, but low ATRX protein abundance. Reduced ATRX protein levels in ALT ATRX wild-type neuroblastoma could be explained by the reduced DAXX protein levels which we specifically observed in this subgroup of tumors. In this scenario, reduced DAXX protein levels impair assembly of the ATRX/DAXX complex, which then causes degradation of orphan ATRX protein molecules. Consistent with this idea, we observed that knocking down DAXX reduced ATRX protein levels in neuroblastoma cells. However, the cause of reduced DAXX protein levels is not yet clear, especially since mRNA levels do not change significantly. Reduced DAXX protein levels in ALT-positive tumors may

result from posttranscriptional events. It is known that DAXX is regulated via various posttranslational modifications including phosphorylation, SUMOylation and ubiquitination⁸. Irrespective of the mechanistic details involved, our study highlights that proteomic data is closer to phenotypes than transcriptomic data, which is especially valuable in a clinical context⁷.” (lines 304 – 322)

4. In Fig. 5b, is SUV39H1 expression altered in ALT+ tumors?

We thank the reviewer for this suggestion. SUV39H1 was unfortunately not detected in the whole proteome analysis. The whole proteome analysis is covering a large amount of proteins, but some proteins are hard to detect and quantify due to their peptide structure, abundance or cellular localization. We added the mRNA expression of *SUV39H1* to Fig. 6. Additionally, we added SUV39H2 mRNA expression. Neither expression of *SUV39H1* nor *SUV39H2* could be directly associated with the observed differences in telomeric H3K9me3 between ALT-positive and ALT-negative tumors.

Figure 6d:

SUV39H1/H2 (only RNA) in ALT tumors compared to tumors from the other groups (TERT, MNA, OTHER).

5. In Fig. 6, is microhomology to telomeric repeats found at 18q23 and 19q13.43?

Yes, microhomology is also found at the hotspot regions on chr18q23 and chr19q13.43. Ten of the twelve events contributing to the hotspot on chr18q23 and twelve of the sixteen events contributing to the hotspot on chr19q13.43 show at least 1 bp microhomology to the canonical TTAGGG sequence. To better illustrate the microhomology of the individual telomeric repeat loci, we added a karyogram with the degree of microhomology for every telomeric repeat locus of the discovery cohort as Supplementary Fig. 15e.

Supplementary Figure 15e:

Degree of microhomology (color coding) and chromosomal location for all telomeric repeat loci in the discovery cohort.

6. All three hotspots of intrachromosomal telomere loci are very close to chromosome ends. What about the intrachromosomal telomeric repeats far from chromosome ends? Do their breakpoints also have microhomology to telomeric repeats? Some of the less frequent telomeric repeat loci are very close to chromosome ends too (on Chr1, some loci are even closer to telomeres than 1q42.2). Do the breakpoints of these loci have less homology to telomeric repeats? What determines the frequency of neo-telomere formation?

Microhomology to the canonical telomeric repeat sequence TTAGGG is a general feature of telomeric repeat loci. Microhomology can only be assessed for telomeric repeat loci showing t-type repeats directly at the junction site. In total, 80.3% (212/264) of all t-type (TTAGGG) containing telomeric repeat loci showed at least 1bp microhomology indicating that telomeric repeat sequences at these loci are added via a microhomology dependent mechanism or via non-homologous end joining¹² (now added as Supplementary Fig.15d). The degree of microhomology for every event is now given in Supplementary Fig.15e (see comment above).

Telomeric repeat loci frequently overlap with chromosomal breakpoints of copy number changes and structural variations (revised Supplementary Fig 15g). Regions of genomic instability and fragile sites are prone to breakage and thus genomic instability of a locus may also be a prerequisite for the occurrence of a telomeric repeat locus. Addition of telomeric repeats to open chromosomal breaks is probably more likely when there is more microhomology to the telomeric repeat sequence. All hotspots for telomeric repeat loci show a high degree of microhomology. Telomeric repeat loci close to the telomere (< 5 MB) do not have a significantly higher degree of microhomology (now added as Supplementary Fig. 15f).

Supplementary Fig. 15:

(d) The occurrence of homologous bases (microhomology) between the reference genome and TTAGGG telomere repeats was counted for all telomeric repeat loci in the discovery cohort. (f) Degree of microhomology to TTAGGG of telomeric repeat loci being less than 5 MB or more than 5 MB away from the chromosome end. (g) (left pie) Percentage of telomeric repeat loci overlapping with a copy number variation (events in group “chr end” are too close to the telomere and thus no CNV information could be obtained). (right pie) Percentage of copy number neutral telomeric repeat loci overlapping with an SV breakpoint within a 10 kb window (TRA = translocation, TEL loci = another telomeric repeat locus, Multiple SVs = more than one SV in 10 kb distance, none = no SV in 10 kb window).

7. For the intrachromosomal telomeric repeats far from chromosome ends, are they flanked by non-telomeric sequences on “both sides”? How would the “neo-telomere” model explain this type of loci? This type of loci seems to increase in ALT_ATRXwt tumors as well. Are they generated through a mechanism different from those hotspots?

In general, almost all telomeric repeat loci were one-sided, meaning that telomeric repeats were present either upstream or downstream of the junction site. At five genomic positions in the discovery cohort, we observed two events in close proximity with opposite orientation of the telomeric repeats. These events were referred to as two-sided (Supplementary Fig. 15a-b). Two-sided events can either result from two independent telomeric repeat loci in close proximity or can form a true telomere insertion meaning that the telomeric sequence is connecting the two junction sites (now discussed in lines 362 - 368). However, evidence for a true insertion (mates of a read pair matching to both sides of the insertion), was only detected for two of these two-sided events (data not shown). Since we were using short read sequencing, the lack of reads supporting a true insertion can also be due to the length of the sequencing reads.

We revised the description of our hypothesis of neo-telomere formation and also included a graphic to illustrate it (Figure 8).

The discussion text now reads (note that reference numbers were adapted to match the order of references in the point-by-point response and differ from the original text):

“ALT-positive neuroblastoma tumors exhibited a high rate of telomeric repeat loci. Since telomeric repeat loci were characterized by telomeric repeat sequences either upstream or downstream of a non-telomeric junction site, these events cannot have occurred from breakage of interstitial telomeric sequences (ITS). However, telomeric repeat loci frequently overlapped with breakpoints of copy number changes or structural variations. Because terminal chromosomal breaks are in need of telomeric repeats to protect the newly formed ends from degradation¹³⁻¹⁵, we propose that ALT-positive tumors are capable of adding telomeric repeat sequences to open ends of chromosomal breaks forming neo-telomeres (Figure 8a). The high degree of microhomology to the telomeric repeat sequence at the junction sites indicates that telomeric repeats are added via a microhomology dependent process like microhomology mediated end joining or non-homologous end-joining¹². Further, we propose that the presence of microhomology at an open chromosomal break determines if telomeric sequences can be added to this site. Chromosomal loss of certain fragments might present a selection advantage leading to a selection of cells harboring the neo-telomere. We also identified a subset of copy number neutral telomeric repeat loci with no associated structural variation. This might be due to the fact that these events are subclonal and were thus not detected by the CNV/SV calling algorithm. Moreover, the detection limit of the used copy number algorithm is 50 kb and thus smaller copy number changes cannot be detected. Two-sided events, defined as two telomeric repeat loci in a 10 kb window with opposite orientation of the telomeric repeats, were very rare. These events may represent insertions of telomeric repeat sequences similar to previously described events¹⁶. Only two of five two-sided events exhibited evidence of a true insertion by mates of a read pair mapping to both sides of the insertion. However, for large insertions the used short read sequencing prevents the detection of supporting reads. Alternatively, two-sided events may result from neo-telomere formation on both sides of a breakpoint (Figure 8b).” (lines 345 – 368)

Supplementary Fig. 15b:
Chromosomal location of one-sided and two-sided telomeric repeat loci of the discovery cohort.

Figure 8

a) one-sided

b) two-sided (rare)

Figure 8: Model of neo-telomere formation

Graphical abstract illustrating the hypothesis of neo-telomere formation in ALT-positive neuroblastomas. (a) The majority of telomeric repeat loci are one-sided. We hypothesize that ALT-positive cells are able to add telomeric sequences to open chromosomal breaks to protect them from degradation. (I) Microhomology to TTAGGG favors the formation of a neo-telomere at the open chromosomal break. Loss of some chromosomal regions might present a selection advantage for the cell and cells with a neo-telomere at the chromosomal breakpoint are selected. (II) Without microhomology, structural rearrangements with other chromosomal arms (grey) can represent an alternative route of protecting open chromosomal breaks. (b) Rare two-sided events can result from insertion of telomeric sequences (I) or from the formation of neo-telomeres on both sides of the breakpoint (II). Small circles represent C-Circles. Chromosomes are shown in grey and telomeric sequences in blue.

8. Fig. 4 suggests that ATRX protein could be down regulated independently of mutations. In Fig. 6, it would be helpful to compare the levels of telomeric repeat loci between ATRX-high and ATRX-low tumors. Loss of ATRX protein may be a better marker for intrachromosomal telomeric repeats.

We thank the reviewer for this suggestion. Indeed, the number of telomeric repeat loci is inversely correlated with ATRX protein expression (Spearman correlation $r = -0.41$, $P = 0.01571$). Since low ATRX protein expression was found to be associated with ALT-activity and a high number of telomeric repeat loci was also found to be associated with ALT-activity, this was expected.

However, no good cut point for ATRX protein expression can be defined to separate samples with a high number of telomeric repeat loci. Therefore, we think that the genetic telomere maintenance classification used in our manuscript is a better way to form subgroups.

Additional Figure 2: ATRX protein expression relative to the number of telomeric repeat loci in the discovery cohort. Color coding indicates TMM classification

Reviewer #3 (Remarks to the Author): ALT expert

The paper entitled “Alternative lengthening of telomeres in childhood neuroblastoma from genome to proteome” by Hartlieb et al. aimed at developing studies that could potentially further document the ALT (Alternative Lengthening of Telomeres) pathway (an alternative mechanism to telomerase reactivation, based on telomeric recombination, that allows tumor cells to maintain functional telomeres) in neuroblastoma, the most common extracranial solid tumor occurring mostly in early childhood.

In neuroblastomas, both the reactivation of telomerase pathway and activation of the ALT pathway are synonymous for dismal outcome, as observed in previous studies, and this occurs in around half the cases. On the other hand, the other half of cases show no sign of reactivation of telomere maintenance (either telomerase or ALT) and these tumors have an excellent outcome, either spontaneously regressing or differentiating into benign ganglioneuromas. Here, the authors focused their studies on further documenting neuroblastoma ALT by analyzing various biological and genetic features from large collections of tumors, always comparing with MYCN-amplified tumors and telomerase-sustained tumors.

The work produced by these authors is really outstanding and exemplary. The authors’ approach is really impressive, because it allowed to produce very large amounts of data using RNA sequencing, whole genome sequencing, ChIP-seq, mass spectrometry for whole proteome analysis, analyses of telomere sequences content, telomere length, ALT activity by C-circle assay, and analyses of epigenetics pathways.

An important question associated with neuroblastoma is to understand why tumors with no sign of telomere maintenance mechanism are more benign than tumors with either telomerase- or ALT-maintained tumors, although it may be obvious that these tumor cells eventually lack the potential to maintain chromosomes stable enough to allow continuous proliferation. On the other hand, it is also important to understand the biological features of ALT tumors compared with those of MYCN- and telomerase-tumors in order to better dictate appropriate therapy. The present study provides us with a pretty good number of new findings concerning neuroblastoma ALT characteristics without however making any major breakthrough in this field. These characteristics may or may not apply to other types of ALT cancers. Nevertheless, no doubt that these new findings will be of a great help to better understand neuroblastoma, but also all the other types of ALT cancers.

We thank the reviewer for his positive assessment and are grateful for his suggestions.

I have a number of remarks and questions, exposed below.

1-The percentage of ALT tumors known to be around 10% in previous studies, and in the present one as well (in the cohort of 718 children), was found here to increase to 47.5% when a particular collection of neuroblastoma tumors, namely one composed of relapsed cases, was screened for ALT. I understand that ALT positivity was associated with older age at diagnosis in these relapsed cases as well as in the screening cohort (lines 72-73). From these 47.5% vs 10%, one can understand that either ALT becomes more frequent as the tumors evolve with time, or, alternatively, that relapse triggers some sort of pathway favoring ALT activation. Do the authors have additional information that could potentially allow to favor one hypothesis or the other? In addition, the authors observed that “relapsed ALT tumors showed increased telomere content compared to the matching primary sample” (lines 122-123). Therefore, ALT products and/or consequences appear to accumulate with age. On the other hand, there is a large number of analyzed relapsed tumors that evolved from MYCN or telomerase tumors (hence the increase of ALT tumors from 10 to 47.5%). C-circle measurement can also provide the intensity of the ALT pathway, as you know. Therefore, it would be very informative to provide the average value for C-circle intensity in the relapsed tumors and compare with that in the 66 CC+ tumors of the screening cohort to see if only those previously ALT+ tumors relapsed to ALT+ have increased

ALT activity (either increased telomere content or C-circle value) or if also telomerase tumors relapsed to ALT+ tumors also have high ALT activity.

Most likely there are multiple factors that contribute to the enrichment of ALT-positive cases in the INFORM relapse cohort. ALT-positive tumors exhibit a protracted clinical course of disease and show molecular signals associated with slow proliferation, indicating a slower growth of these tumors. The current treatment protocols for high-risk neuroblastomas are based on a multimodal chemotherapy regimen targeting strongly proliferating cells, including drugs like doxorubicin, etoposide and cisplatin. We propose that this treatment protocol is not well-suited for ALT-positive tumors. Therefore, ALT-positive tumor cells most likely survive the first-line treatment due to their slow growth. Additionally, activation of ALT could be one way to escape the selection pressure of the applied treatment and, thus, represent a therapy resistance mechanism. However, one also has to consider that the inclusion criteria of the INFORM registry include a life expectancy of at least three months and sufficient general condition (Lansky score ≥ 50 or Karnofsky score ≥ 50), which might lead to the exclusion of the most aggressive and fastest progressing tumors. These extremely fast progressing neuroblastomas are rather *MYCN*-amplified or *TERT*-rearranged and thus these cases might to a certain extent be underrepresented. We now discuss this in the revised manuscript (lines 379 - 392).

We thank the reviewer for the suggestion to extend the comparison of telomere content and C-Circle intensity between the two cohorts. We summarized telomere content and C-Circle intensities of neuroblastomas in the discovery and INFORM cohort in Supplementary Fig. 1. Additionally, we included the raw C-Circle images of these cohorts (Supplementary Fig. 2). Overall, telomere content and C-Circle intensity of the discovery cohort and INFORM cohort were comparable. Interestingly, ALT-positive tumors with a very high telomere content (>1.91) had a significantly shorter event-free survival probability (Supplementary Fig. 3d). To gain a better understanding how telomere content and C-Circle intensity evolve over disease progression, we extended the analysis of matching primary and relapse pairs (Supplementary Fig. 4). The extended analysis of primary/relapse pairs did not confirm our previous statement that “relapsed ALT tumors showed increased telomere content compared to the matching primary sample” and on average there was no clear increase in the telomere content between primary and relapse pairs. Thus, we adapted the previous statement in the revised manuscript (line 119 - 120). However, for 3/14 pairs the primary tumor was C-Circle negative, while the relapsed tumor was C-Circle positive, indicating a gain of ALT activity. All three pairs showed a slight increase in telomere content. Two of these three pairs were *MYCN*-amplified in the primary and heterogeneous MNA/ALT in the relapse.

Supplementary Figure 1: Telomere content and C-Circle intensity of discovery and INFORM cohort

(a) Relative telomere content of C-Circle (CC) positive and negative tumors in the discovery and INFORM cohort. (b) C-Circle intensity relative to CHLA-90 positive control for tumors in the discovery and INFORM cohort. Color coding indicates the C-Circle status using the two applied criteria of a signal intensity relative to CHLA-90 positive control ≥ 0.2 and a signal intensity relative to the negative control without polymerase ≥ 4 . Due to the high number of samples, not all tumors could be analyzed on the same blot. Thus, every blot contained a CHLA-90 positive control and signal intensities were normalized to this control. (a-b) P values were calculated using Wilcoxon rank sum tests. (c) Scatter plot of telomere content relative to C-Circle intensity. Color coding indicates C-Circle status.

Supplementary Figure 3d:

Event free survival of ALT-positive neuroblastomas in the discovery cohort. ALT-positive tumors are separated based on a very high (> 1.91) telomere content. Cut point was calculated using Maximally Selected Rank Statistics. P value was calculated using a Log-rank test.

Supplementary Figure 4: Comparison of matching primary and relapse pairs

(a) Telomere content determined using lcWGS data of matching primary and relapse pairs. C-Circle status and MYCN status is given. For samples with available hcWGS data TERT status and ATRX mutation status is given. (b) C-Circle images of matching primary and relapse pairs. Color coding on the left side indicates the TMM group (see legend a). For every tumor, a control without polymerase is shown. For samples labeled with * primary and relapse tumor were not analyzed on the same blot.

2- Lines 210-213: Concerning the enrichment of ALT activity in relapsed tumors, the authors suggest that the unfavorable outcome of these tumors might be due to a “primary resistance to the current standard treatment regimens targeting strongly proliferating tumors, which are probably not suited to treat slowly growing ALT tumors, particularly those with canonical activating RAS pathway mutations”.

It would be important to analyze the treatment received case by case by the children of the INFORM cohort. Indeed, if an inappropriate treatment, namely against strongly proliferating

tumors, was applied to ALT tumors, it will show up in these numbers, because it is known whether the initial status of the relapsed tumors was ALT, or telomerase, or MYCN or no TMM. In fact, I do not understand how treatment may have been ill appropriate at first diagnosis, because, initially, ALT tumors were recognized as such and were probably not treated with drugs targeting strongly proliferating tumors, unless inappropriateness of the treatment was not recognized at the moment. If inappropriateness of the treatment was known, then it was not likely applied and, therefore, the unfavorable outcome of ALT tumors cannot be due to non appropriate treatment.

We apologize for not precisely explaining the current standard of care in Germany. All INFORM cases received first-line treatment based on the NB2004 German Neuroblastoma trial protocol. Risk stratification is done based on stage, patient age, amplified *MYCN* and 1p status. Apart from amplified *MYCN*, the presence of a telomere maintenance mechanism (*TERT*-rearrangement or ALT-activity) is not considered in the risk stratification. A precise TMM classification is not done at first diagnosis and the TMM status of the matching primary tumor is only known for a small subset of matching pairs analyzed as part of this study. Based on the risk stratification into low risk, intermediate risk and high risk, different treatment protocols are applied. According to this risk stratification, ALT-positive neuroblastomas classified as high risk (based on stage and age) will be treated with a multimodal chemotherapy protocol, including classical chemotherapeutic drugs targeting fast proliferating cells like cisplatin, doxorubicin and etoposide. ALT-positive neuroblastomas classified as low-risk, will be monitored with a wait-and-see strategy and intermediate risk tumors will be treated with chemotherapy (see NB2004 trial protocol). We now explain this in more detail in line 72-79. The study by Ackermann et al.¹⁰ could show that the presence of an active telomere maintenance mechanism is an indicator of poor survival independent of the current risk stratification and proposed a revised molecular risk stratification including telomere maintenance status. However, this revised concept is so far not applied in the clinic.

3- Lines 76-77: Do you know why the “Amplified MYCN and C-Circle presence were mutually exclusive in the screening/discovery cohort, but not in the INFORM cohort (Fig. 1b)”? Since the MYCN amplified C-circle positive INFORM tumors concern only two patients (out of the 19/40 positive ones), the significance of this absence of mutual exclusion between MYCN amp and C-circle + may not be really due to a well defined molecular event. Have the authors thought of a possible explanation for this? If MYCN-amp and ALT are really present together in these two tumors, can these identify a particular subtype of neuroblastoma in which ALT and telomerase activation could co-exist?

To further analyze the heterogeneous cases, we included telomere content analysis, *MYCN* status and C-Circle status of the matching primary tumors of NBI5 and NBI26 (Supplementary Fig. 4). Both primary tumors were C-Circle negative and *MYCN*-amplified and had a log₂ telomere content tumor/control ratio below zero, which is typical for *MYCN*-amplified cases. Telomere content was slightly elevated in the relapse disease period compared to the primary. Presence of both amplified *MYCN* and ultra-bright telomere spots characteristic for ALT-positive cells were identified in FISH analysis of NBI5 (now added as Supplementary Fig. 5a). Unfortunately, for NBI26 there was no tumor material left for FISH analysis. However, FISH analysis of the matching primary of NBI26 using a telomere probe and a *MYCN* probe could show evidence of an ALT-positive subclone (now added as Supplementary Fig. 5b).

Our discovery cohort also contained a few heterogeneous cases (HET ALT/MNA). These HET cases in the discovery cohort were C-Circle negative using our criteria for C-Circle assessment, but exhibited a very high telomere content and thus were considered HET ALT/MNA. FISH analysis using a telomere and *MYCN* probe of one of these heterogeneous cases could show that both *MYCN* amplification and ultra-bright telomere spots were present in the same tumor

(now added as Supplementary Fig. 5c). For all further analysis, the group of HET cases was always kept separate, since these cases are not fully understood. HET cases might only display certain features of ALT-activity or the C-Circle level might be too weak to be detected as positive, since C-Circles might only be present in a subset of tumor cells. Indeed, coexistence of ALT markers and amplified *MYCN* is possible, but rather rare and heterogeneous cases should be considered a special subgroup.

Supplementary Figure 5: Combined *MYCN* and telomere FISH

(a) *MYCN* and telomere FISH of HET MNA/ALT tumor NBI5. Additionally, DAPI staining and a merged image are shown. Scale bar representing 20 μm . (b) PML immunostaining and telomere FISH of the matching primary of HET MNA/ALT NBI26 (top panel). *MYCN* and CEP2 FISH of the matching primary of HET MNA/ALT tumor NBI26 (middle panel). *MYCN* and telomere FISH of the matching primary of HET MNA/ALT tumor NBI26 (bottom panel). Scale bar representing 10 μm . (c) *MYCN* and telomere FISH of HET MNA/ALT tumor NBD151. Additionally, DAPI staining and a merged image are shown. Scale bar representing 20 μm .

4- Lines 152-154: “Interestingly, ALT-positive ATRX wild-type tumors exhibited reduced protein levels of the ATRX interaction partner DAXX (Fig. 4c), pointing towards an alternative route of loss of ATRX/DAXX complex activity”. This was observed in all 5 ATRX-wt ALT+ tumors tested. Do you think that this observation can now allow to conclude that there is no other type of ALT tumor (in neuroblastoma at least) besides the ATRX-mutated tumors (55%) and the ATRX-wt DAXX low expressed tumors and that ALT always results from ATRX-DAXX dysfunction? If yes, this would be very informative to other researchers willing to check this in other types of ALT cancers.

We thank the reviewer for this comment. We propose that apart from ALT-positive neuroblastomas with mutated *ATRX*, there is a second subgroup of ALT-positive neuroblastomas, which exhibits reduced ATRX/DAXX complex expression. We observed that all five *ATRX* wild-type tumors tested showed reduced DAXX protein levels. It is tempting to speculate that all ALT-positive tumors fall into two categories – *ATRX*-mutated tumors or *ATRX* wild-type tumors with reduced DAXX protein levels. However, we don't think the number of tumors we analyzed is large enough to draw this general conclusion. We fully agree with the reviewer that this should be validated in larger cohorts and potentially other ALT-positive entities. The overall scarcity of tissue form ALT-positive tumors represents a significant challenge for ongoing and future studies.

5- Lines 92-93 and Suppl. Fig. 2: Have the authors examined in even much more detail the presence of mutations in the ALT+-*ATRX*-mut versus ALT+-*ATRX*-wt tumors and, more precisely, analyzed possible correlations between the two groups of tumors? I'll explain. As I understand, the around 45% of ALT-*ATRX*-wt tumors of the discovery cohort suffer from very affected DAXX levels (5/5, Fig. 4c). Therefore, one might think, as I said in my precedent point and as you suggest, that in these ALT+-*ATRX*-wt tumors, the reason for the presence of ALT activity is also due to the absence of a functional ATRX-DAXX complex and, therefore, all types of ALT tumors would have the same origin, a loss of functional ATRX-DAXX complex activity. My question is: If the reason for the existence of ALT activity is only due to loss of either ATRX or DAXX, we should expect all other mutations, shown in Suppl. Fig. 2, to be rather similar in the two groups, which is not the case. How can we explain that? And, in addition, have you looked whether these differences in mutations between the two groups could give clues to understand the origin of the decrease in DAXX levels in these tumors? Or, else, to understand the type of post-transcriptional regulation of ATRX you suggest (based on actual observations)? On the other hand, these 5 cases of decreased DAXX levels might not be numerous enough to extrapolate and conclude that all ALT+-ALT-WT tumors are ALT positive because of reduced DAXX levels? Or, else, the extent of decrease might not be large enough to provoke total loss of function of the ATRX-DAXX complex?

Again, we greatly appreciate the thoughtful comments by the reviewer. First, we indeed think that loss of ATRX/DAXX complex activity may be the universal molecular feature underlying the ALT phenotype. However, as outlined in our reply to point 4 and as also pointed out by the reviewer, we do not have sufficient data to draw a firm conclusion. By adding this point to the discussion section (lines 304 - 322), we think that we will stimulate further investigation of this question in neuroblastoma and other types of cancer (see our reply to point 4).

ATRX-mutated cases have a higher total SV count, which could indicate a higher genetic instability in these tumors. Overall, the mutations between *ATRX*-mutated and wild-type tumors are rather similar. This is depicted in the right panel heatmap of new Supplementary Fig. 6 (old Suppl. Fig. 2) and Supplementary 7c, which is giving the frequency of mutations (SVs, SNVs, INDELS) in the different TMM groups (ALT_*ATRX*mut, ALT_*ATRX*wt, MNA, TERT and OTHER). Additionally, we searched for recurrent mutations in ALT-positive *ATRX* wild-type tumors (Supplementary Fig. 12b) and mutations in ATRX and DAXX interaction partners (according to

the BioGrid database; Supplementary Fig. 13) to potentially identify events contributing to loss of ATRX/DAXX. No recurrent mutation patterns in ATRX/DAXX interaction partners could be identified. Furthermore, changes in DAXX protein expression are not reflected on the mRNA level. Together this indicated that DAXX expression is regulated on the posttranscriptional level. Interestingly, DAXX is described in the literature to be subject to various post-translational modifications and is also regulated by ubiquitin-mediated degradation⁸ (manuscript lines 318 - 322).

If ATRX mutation and low ATRX/DAXX protein abundance is leading to the same extend to loss of ATRX function, would be a very interesting question. However, to experimentally address this question would be beyond the scope of this manuscript.

6- Lines 100-101: You stated that overall survival was not significantly different between ATRX-mutated and ATRX wild-type ALT cases (Fig. 3c-d). Yet, in Fig. 3c, the survival of the ALT+-ATRX-wt patients after ~ 12 years seems to be much more than that in the ALT+-ATRX-mut patients. None of the latter group survived after ~ 14 years, unlike in the former group.

To better illustrate the number of samples that are still observed/alive at a certain time point, we added a table with the number of patients at risk below the survival curve. (We also included the table for all other survival curves in the paper.). Yes, it is correct that after 15 years, none of the ALT ATRX-mutated cases is still alive, while four of the ALT ATRX wild-type cases are still alive. However, four cases are not enough to draw any reliable conclusions and for three out of these four cases the observation time ends shortly after the 15 year time point. Taken together, it might be possible that ALT-positive ATRX wild-type neuroblastomas have a slightly better overall survival probability, but more samples per group and a longer observation time for the individual patients would be necessary to draw meaningful conclusions.

Figure 3e:

Event-free and overall survival rate of patients having an ATRX-mutated ALT-positive neuroblastoma ($n = 33$) compared to patients with an ATRX wild-type ALT-positive neuroblastoma ($n = 27$). P values were calculated using a log rank test.

7- It was surprising to learn that POLD3 was mutated in a substantial number (50%) of ALT tumors (Suppl. Fig. 4), as POLD3 has recently been reported to be essential for ALT in human tumors (Dilley et al, Nature, 539 (2016) 54-58; Roumelioti et al, EMBO Rep, 17 (2016) 1731-1737). Had the authors noticed this point? And would they like to comment on it? In addition, this POLD3 mutation is substantiated by the fact that you also found by mass spec

that POLD1, another subunit of DNA polymerase delta had diminished levels (entry #182 of Suppl. Table 4), as had POLDIP3, a POLD3-interacting protein (entry #253).

We thank the reviewer for this suggestion. Copy number loss of *POLD3* is frequently observed in ALT-positive neuroblastomas (now Supplementary Fig. 7c). Overall, *POLD3* mRNA expression is also lower in tumors exhibiting a *POLD3* loss (Additional Fig. 3). However, *POLD3* is located on Chr11q, which is frequently lost in non-*MYCN*-amplified neuroblastomas^{17,18} including ALT-positive tumors (Supplementary Fig. 8). Loss of *POLD3*, which is in this case part of a larger chromosomal loss, is not only seen in ALT-positive neuroblastomas, but also frequent in other neuroblastomas without amplified *MYCN*. The remaining arm of chr11q might be sufficient to fulfill the ALT-essential role of POLD3 described in the literature. Similar to the frequent loss of *POLD3* due to chr11q loss, we observed frequent loss of *ATM*, which is also located on chr11q.

To make this clearer we added the following text to the revised manuscript (note that reference numbers were adapted to match the order of references in the point-by-point response and differ from the original text):

“Copy number loss of POLD3 and ATM was frequently observed in ALT-positive tumors (Supplementary Fig. 7c). However, both POLD3 and ATM are located on chr11q and loss of these genes was associated with loss of chr11q, which is frequently observed in ALT-positive (Supplementary Fig. 8) and other non-MYCN-amplified neuroblastomas^{17,18}.” (lines 147 – 151)

Additional Figure 3: POLD3 mRNA expression in the discovery cohort.
Color coding indicates CNVs affecting POLD3.

8- There is an apparent inverse correlation between the frequency of telomeric repeat loci and the protein level of EXO1. Indeed, the telomeric repeat frequency was significantly higher in the ALT+-ATRX-mut tumors than in the ALT+-ATRX-WT tumors (Fig. 6a), while, on the other hand, EXO1 protein levels were lower in the ALT+-ATRX-mut tumors than in the ALT+-ATRX-WT tumors (Suppl. Fig. 10). Could this correlation be real? And if yes (but how to verify this?), how could it be explained?

Perhaps I did not completely understand the mechanistic of telomere repeat insertion, but in my mind, loss of a chromosome fragment (such as that containing EXO1 at 1q42.2) will always correspond to one telomere repeat Insertion event and, therefore, the increased frequency of

telomere repeat in ALT+-ATRX-mut tumors cannot in itself explain the diminution of EXO1 expression in these tumors.

Indeed, EXO1 protein expression is inversely correlated with the number of telomeric repeat loci (Spearman correlation $r = -0.55$, $P = 0.00067$; Additional Fig. 4). However, there is no significant correlation on the mRNA level (Spearman correlation, $r = -0.15$, $P = 0.076$; Additional Fig. 4). EXO1 protein expression is not significantly different between ALT-positive ATRX-mutated and ALT-positive ATRX wild-type tumors ($P = 0.43$, Supplementary Fig. 16c). A high number of telomeric repeat loci was found to be associated with ALT-activity in neuroblastoma tumors. Low EXO1 protein and mRNA expression was also found to be associated with ALT-activity. Therefore, we do not believe that the correlation of EXO1 expression and the number of telomeric repeat loci is biologically relevant. EXO1 is one candidate gene located on chr1q43, which is affected by the 1q42-1qter deletions associated with a telomeric repeat locus on 1q42 seen in some ALT-positive neuroblastomas. We hypothesize that the recurrence of this event indicates a selective advantage of this deletion. Loss of chr1q42 is only seen in a subset of ALT-positive neuroblastomas and the fact that EXO1 expression is in general low in these tumors could indicate that there are alternative ways of silencing this gene.

Since we possibly did not explain our hypothesis and the concept of neo-telomere formation precisely enough, we revised this text paragraph (lines 345 - 368) and added a graphical abstract/cartoon (Figure 8) for our hypothesis on neo-telomere formation in neuroblastoma.

Additional Figure 4: EXO1 expression relative to number of telomeric repeat loci

EXO1 protein (left) and mRNA (right) expression relative to the number of telomeric repeat loci in a tumor. Color coding indicates the TMM group

Reviewer #4 (Remarks to the Author): Proteomics expert

Hartlieb and colleagues use multi-omics approaches to comprehend the molecular features characterizing neuroblastomas, in particular ALT positive tumors from large patient cohorts. Unlike telomerase, it is difficult to 'quantify' ALT activity, if that is a biologically relevant thing to do. Here the authors score ALT in neuroblastomas by detecting C-Circles, one of the most reliable, but not definitive, ALT marker. They perform this on a screening cohort (n=718 tumors) and on the INFORM cohort (n=40). They performed whole genome sequencing, transcriptomics, CHIP sequencing and whole proteome analysis, and looked for correlations between ALT (C-circle presence) and their omics dataset and clinical data. From these data they confirm that ALT neuroblastomas are biologically distinct from telomerase positive tumors. The main observed characteristics from the ALT tumors are the following:

- (i) higher telomeric DNA content than in ALT negative tumors
- (ii) low TERT RNA expression, TERRA accumulation
- (iii) a complex mutational landscape. A rather high ATRX mutation rate, DAXX and H3F3A mutations are rare. Importantly, ATRX wt tumors can display ALT features, however, it is unclear if ATRX function is 'normal' in these cases: e.g. some cases show normal ATRX mRNA expression but low ATRX protein levels.
- (iv) telomeric H3K9me3 enrichment and H3K27me3 depletion
- (v) high frequency of ALT specific interstitial telomeric repeats which are unstable, and identification of recurrent telomeric amplified repeat hotspots, for example at the 1q42.2 chromosomal position.

The study is of high quality and very interesting because of the size of the screening cohort and the inclusion of relapsed tumors (INFORM). The study is essentially confirmatory and descriptive for the most parts. Nonetheless, I think it will be useful for researchers in the fields of cancer and telomere biology. Therefore, I am supporting the publication of this article, pending many minor revisions.

We are very grateful for the positive feedback and the constructive comments to improve our manuscript.

(1-) Raw C-Circle blots are missing. Would it be possible to show these, at least for a subset of representative tumors?

We added the raw C-Circle blot images of the neuroblastomas in the discovery cohort and INFORM cohort as new Supplementary Fig. 2.

(2) Telomerase assays, which are quite straightforward, should be attempted also on a subset of representative tumors

We added telomerase activity assays for a subset of the discovery cohort to Supplementary Fig. 3b. ALT-positive tumors had minimal telomerase activity, while *TERT*-rearranged and *MYCN* amplified tumors have significantly higher telomerase activity.

The revised text now reads:

"In accordance with low TERT mRNA expression, ALT-positive tumors exhibited low telomerase activity (Supplementary Fig. 3b)." (Lines 108 - 109)

Supplementary Figure 3b:
Relative telomerase activity of a subset of neuroblastomas of the discovery cohort.

(3) Previous published data have shown that ALT is not present in MYCN-amplified tumors, and that MYCN amplification and ATRX mutation are generally exclusive. How do the authors explain the co-occurrence of MYCN amplification with some ALT markers (detected C-Circles and a slight increase of telomere content but ATRXwt) in the INFORM cohort (NBI5-NBI26)? Could it be explained by the acquisition of one or the other TMM at the relapse? Can it be ALT independent? For instance, replication stress and trimming dysfunction can generate extrachromosomal Circle accumulation independently of the ALT pathway.

Although these cases are rather rare, we do see some co-occurrence of MYCN amplification and ALT markers. FISH analysis of NBI5 using a telomere and MYCN probe showed that ALT-specific ultra-bright telomere spots and amplified MYCN co-occur in this tumor (now added as Supplementary Fig. 5a). Additionally, in the matching primary of NBI26, we found evidence of an ALT-positive subclone using FISH analysis (now added as Supplementary Fig. 5b). Both HET ALT/MNA cases of the INFORM cohort were MYCN-amplified and C-Circle negative at diagnosis. We additionally added telomere content analysis of these matching pairs and there was a slight increase of the telomere content from primary to relapse for these heterogeneous pairs (Supplementary Fig. 4a).

Our discovery cohort also contains three HET ALT/MNA cases with very high telomere content and amplified MYCN. FISH analysis using a MYCN and a telomere probe could also confirm heterogeneity in NBI151 (now shown in Supplementary Fig 5c).

Interestingly, none of the heterogeneous MNA/ALT cases had an ATRX mutation (for NBI26 only lcWGS and WES data was available). Overall, the number of these heterogeneous cases is very low and thus it is hard to draw general conclusions. Due to the limitation of tumor material, it was not possible to look at telomerase activity in these tumors to evaluate if both telomere maintenance mechanisms are active. Heterogeneity of ALT and amplified MYCN is discussed in the revised manuscript text in lines 323 – 335.

Supplementary Figure 5: Combined MYCN and telomere FISH

(a) MYCN and telomere FISH of HET MNA/ALT tumor NBI5. Additionally, DAPI staining and a merged image are shown. Scale bar representing 20 μ m. (b) PML immunostaining and telomere FISH of the matching primary of HET MNA/ALT NBI26 (top panel). MYCN and CEP2 FISH of the matching primary of HET MNA/ALT tumor NBI26 (middle panel). MYCN and telomere FISH of the matching primary of HET MNA/ALT tumor NBI26 (bottom panel). Scale bar representing 10 μ m. (c) MYCN and telomere FISH of HET MNA/ALT tumor NBD151. Additionally, DAPI staining and a merged image are shown. Scale bar representing 20 μ m.

(5) “reduced ATRX-DAXX complex activity” (l.47), “loss of ATRX/DAXX complex activity” (l.153), “reduced protein complex activity” (l.205): no experiment has been performed to measure activity or the genome wide binding of these proteins. The authors only observe decreased protein levels by LFQ proteomics, they should avoid these inappropriate statements.

We apologize for this overstatement. The term ATRX/DAXX complex activity was replaced by ATRX/DAXX complex abundance in the revised manuscript text.

(6) The parameters used for ChIPseq telomeric sequences searching by TelomereHunter are not available in the material and methods.

We apologize for missing this part in our methods description. We added the necessary information in the revised version.

(7) H3K9me3 enrichment, H3K27me3/Ac depletions at ALT telomeres. Sometimes the log2 enrichment values are below zero, which indicate very weak signals. But the problem with this representation is that one cannot estimate if the telomeres are truly enriched or depleted of these marks compared to other known loci (unique or repeated) where these marks are known to be present: like pericentromeric regions for H3K9me3 (alpha-satellites), or the telomerase promoter for H3K27me3 (values that you have reported in another figure). H3K27ac and H3K27me3 enrichments at satellite alpha and III are also a good negative control and give an idea of the background than histone ChIP sometimes have. The authors should add these analyses.

We appreciate the suggestion of Reviewer #4 to include additional control analysis of our ChIP-seq data. In accordance with a publication by Cubiles et al.¹⁹ we included the enrichment values of the canonical sequence of alpha-satellite SatII and SatIII. This analysis was included as Supplementary Fig. 14. Confirming previous observations^{19,20}, SatII and SatIII sequences were enriched for H3K9me3, while H3K27ac, H3K27me3 and H3K36me3 were not enriched (Supplementary Fig. 14a).

The level of H3K9me3 at the telomeres of ALT-positive neuroblastomas was comparable with the levels of H3K9me3 at the SatII and SatIII, which is substantiating that the telomeres of ALT-positive neuroblastomas are enriched for H3K9me3 (Supplementary Fig. 14b).

H3K27ac levels were overall rather low at telomeres, SatII and SatIII sequences. However, the levels in non-ALT tumors were higher at all observed sites, indicating that there might be a general difference in H3K27ac, which is not telomere specific (Supplementary Fig. 14c). H3K27me3 levels at the telomeres of ALT-positive neuroblastomas were very low and comparable to the level of H3K27me3 at SatII and SatIII (Supplementary Fig. 14d). Low H3K36me3 was observed at the telomeres, SatII and SatIII sequences of both ALT-positive and ALT-negative tumors (Supplementary Fig. 14e).

Supplementary Figure 14: Chromatin state at SatII and SatIII sequences

(a) log₂ enrichment of ChIP signals relative to input signals of SatII and SatIII sequences for H3K9me₃, H3K27ac, H3K27me₃ and H3K36me₃. ALT-positive tumors were compared to ALT-negative tumors (b-e) log₂ enrichment in ChIP of H3K9me₃ (b, n = 26), H3K27ac (c, n = 25), H3K27me₃ (d, n = 25) and H3K36me₃ (e, n = 27) relative to input. Enrichment at telomeres, SatII and SatIII sequences is given. Additionally, telomeric signals were normalized to either SatII or SatIII. ALT-positive tumors are compared to ALT-negative tumors.

(8) (l.125-128) Unclear: In Sup. Fig.6a the authors present data from the Inform cohort (NBI) and not from the discovery cohort (NBD). However, in the sentence they compare NBI and NBD for RAS mutation. Moreover, the word “supporting a specific impact of aberrant RAS signaling in relapsed ALT positive tumors” is not appropriate, no biological experiments have been done to confirm dysfunctional RAS signaling in those tumors.

We apologize for missing the correct reference to Supplementary Fig. 10 (old 6) and Fig. 3a, since the statement is referring to a difference in the frequency of RAS pathway mutations in ALT-positive tumors between the discovery and INFORM cohort. To better illustrate this comparison we added Supplementary Fig. 10b in the revised manuscript.

Furthermore, the statement “supporting a specific impact of aberrant RAS signaling in relapsed ALT-positive tumors” was changed to “supporting a specific impact of RAS pathways mutations in relapsed ALT-positive tumors” in the revised manuscript text. We apologize for this overstatement.

Supplementary Figure 10b:

Frequency of RAS pathway mutations (HRAS, KRAS, NRAS, BRAF, RAF1, NF1, CDK4, CCND1) in ALT-positive neuroblastomas in the discovery and INFORM cohort. Relapse cases in the discovery cohort and HET cases in the INFORM cohort were excluded from the analysis.

(9) “many proliferation-associated proteins” (l.143), the authors should be more precise, how many? Which pathways? not clear in the Sup. Fig.7a.

We thank the reviewer for this suggestion. To extend the description of the proteomics part, Supplementary Fig 7a was moved to the main Figure 4b. Additionally, we included the proteins that contribute to the terms in Figure 4b as Supplementary Data 6.

The revised text now reads (note that reference numbers were adapted to match the order of references in the point-by-point response and differ from the original text):

“Many pathways associated with proliferation were significantly downregulated on protein level in ALT-positive tumors compared to the telomerase-activated tumors including “DNA replication” and “chromosome”. Notably, the classical proliferation marker MKI67¹, comprised in the term “chromosome”, was one of the top significantly downregulated proteins in ALT-positive neuroblastomas (Fig. 4a-b, Supplementary Fig. 11b). The term DNA replication includes among others the proteins PCNA, MCM2, MCM3, MCM4, MCM5 and MCM7, which are also indicative of the proliferative capacity of a cell². Furthermore, the term “bromodomain” was significantly reduced in ALT-positive tumors, while strongly upregulated in MNA tumors. Amplified MYCN is known to be associated with increased occupancy of active promoter regions and enhancer invasion by MYCN and bromodomain proteins, leading to an increased transcriptional activity of many proliferation associated genes and downregulation of differentiation genes³. Moreover, ALT-positive tumors exhibit a significantly lower fraction of cycling cells compared to MYCN-amplified tumors (Supplementary Fig. 11c). Taken together, both protein expression data and cell

cycle analysis support a low proliferative capacity of ALT-positive neuroblastomas.” (lines 180 – 194)

(10) The authors conclude “ALT tumors exhibited a high rate of neo-telomere formation” (l.207) There is no experimental data showing this conclusion. They suggest this hypothesis before in the text, but the loss of chromosomal copy number correlated with telomeric insertion could be also the result of chromosome internal ITS instability, these interstitial repeats may promote fragile hotspots prone to breakage in ALT cells.

Wording, what is the hypothesis, what is actually proven, what is a neo-telomere?

We apologize for not being accurate with the separation of observation and hypothesis. To better explain our hypothesis of a frequent neo-telomere formation in ALT-positive neuroblastomas, we added a graphical abstract as Figure 8. To better separate our observations from our hypothesis/model of neo-telomere formation, the description of the neo-telomere model was moved to the discussion of the revised manuscript.

The revised text now reads (note that reference numbers were adapted to match the order of references in the point-by-point response and differ from the original text):

“ALT-positive neuroblastoma tumors exhibited a high rate of telomeric repeat loci. Since telomeric repeat loci were characterized by telomeric repeat sequences either upstream or downstream of a non-telomeric junction site, these events cannot have occurred from breakage of interstitial telomeric sequences (ITS). However, telomeric repeat loci frequently overlapped with breakpoints of copy number changes or structural variations. Because terminal chromosomal breaks are in need of telomeric repeats to protect the newly formed ends from degradation¹³⁻¹⁵, we propose that ALT-positive tumors are capable of adding telomeric repeat sequences to open ends of chromosomal breaks forming neo-telomeres (Figure 8a). The high degree of microhomology to the telomeric repeat sequence at the junction sites indicates that telomeric repeats are added via a microhomology dependent process like microhomology mediated end joining or non-homologous end-joining¹². Further, we propose that the presence of microhomology at an open chromosomal break determines if telomeric sequences can be added to this site. Chromosomal loss of certain fragments might present a selection advantage leading to a selection of cells harboring the neo-telomere. We also identified a subset of copy number neutral telomeric repeat loci with no associated structural variation. This might be due to the fact that these events are subclonal and were thus not detected by the CNV/SV calling algorithm. Moreover, the detection limit of the used copy number algorithm is 50 kb and thus smaller copy number changes cannot be detected. Two-sided events, defined as two telomeric repeat loci in a 10 kb window with opposite orientation of the telomeric repeats, were very rare. These events may represent insertions of telomeric repeat sequences similar to previously described events¹⁶. Only two of five two-sided events exhibited evidence of a true insertion by mates of a read pair mapping to both sides of the insertion. However, for large insertions the used short read sequencing prevents the detection of supporting reads. Alternatively, two-sided events may result from neo-telomere formation on both sides of a breakpoint (Figure 8b).” (lines 345 – 368)

Figure 8: Model of neo-telomere formation

Graphical abstract illustrating the hypothesis of neo-telomere formation in ALT-positive neuroblastomas. (a) The majority of telomeric repeat loci are one-sided. We hypothesize that ALT-positive cells are able to add telomeric sequences to open chromosomal breaks to protect them from degradation. (I) Microhomology to TTAGGG favors the formation of a neo-telomere at the open chromosomal break. Loss of some chromosomal regions might present a selection advantage for the cell and cells with a neo-telomere at the chromosomal breakpoint are selected. (II) Without microhomology, structural rearrangements with other chromosomal arms (grey) can represent an alternative route of protecting open chromosomal breaks. (b) Rare two-sided events can result from insertion of telomeric sequences (I) or from the formation of neo-telomeres on both sides of the breakpoint (II). Small circles represent C-Circles. Chromosomes are shown in grey and telomeric sequences in blue.

- 1 Perou, C. M. *et al.* Distinctive gene expression patterns in human mammary epithelial cells and breast cancers. *Proceedings of the National Academy of Sciences of the United States of America* **96**, 9212-9217, doi:10.1073/pnas.96.16.9212 (1999).
- 2 Whitfield, M. L., George, L. K., Grant, G. D. & Perou, C. M. Common markers of proliferation. *Nature reviews. Cancer* **6**, 99-106, doi:10.1038/nrc1802 (2006).
- 3 Zeid, R. *et al.* Enhancer invasion shapes MYCN-dependent transcriptional amplification in neuroblastoma. *Nature genetics* **50**, 515-523, doi:10.1038/s41588-018-0044-9 (2018).
- 4 Braun, D. M., Chung, I., Kepper, N., Deeg, K. I. & Rippe, K. TelNet - a database for human and yeast genes involved in telomere maintenance. *BMC Genet* **19**, 32, doi:10.1186/s12863-018-0617-8 (2018).
- 5 Juskiewicz, S. & Hegde, R. S. Quality Control of Orphaned Proteins. *Molecular cell* **71**, 443-457, doi:10.1016/j.molcel.2018.07.001 (2018).
- 6 Taggart, J. C., Zaubner, H., Selbach, M., Li, G. W. & McShane, E. Keeping the Proportions of Protein Complex Components in Check. *Cell Syst* **10**, 125-132, doi:10.1016/j.cels.2020.01.004 (2020).
- 7 Buccitelli, C. & Selbach, M. mRNAs, proteins and the emerging principles of gene expression control. *Nature reviews. Genetics*, doi:10.1038/s41576-020-0258-4 (2020).
- 8 Mahmud, I. & Liao, D. DAXX in cancer: phenomena, processes, mechanisms and regulation. *Nucleic acids research* **47**, 7734-7752, doi:10.1093/nar/gkz634 (2019).
- 9 Azzalin, C. M. & Lingner, J. Telomere functions grounding on TERRA firma. *Trends Cell Biol* **25**, 29-36, doi:10.1016/j.tcb.2014.08.007 (2015).
- 10 Ackermann, S. *et al.* A mechanistic classification of clinical phenotypes in neuroblastoma. *Science* **362**, 1165-1170, doi:10.1126/science.aat6768 (2018).
- 11 Peifer, M. *et al.* Telomerase activation by genomic rearrangements in high-risk neuroblastoma. *Nature* **526**, 700-704, doi:10.1038/nature14980 (2015).
- 12 Ottaviani, D., LeCain, M. & Sheer, D. The role of microhomology in genomic structural variation. *Trends Genet* **30**, 85-94, doi:10.1016/j.tig.2014.01.001 (2014).
- 13 Guilherme, R. S. *et al.* Terminal 18q deletions are stabilized by neotelomeres. *Mol Cytogenet* **8**, 32, doi:10.1186/s13039-015-0135-6 (2015).
- 14 Flint, J. *et al.* Healing of broken human chromosomes by the addition of telomeric repeats. *American journal of human genetics* **55**, 505-512 (1994).
- 15 Wilkie, A. O., Lamb, J., Harris, P. C., Finney, R. D. & Higgs, D. R. A truncated human chromosome 16 associated with alpha thalassaemia is stabilized by addition of telomeric repeat (TTAGGG)_n. *Nature* **346**, 868-871, doi:10.1038/346868a0 (1990).
- 16 Marzec, P. *et al.* Nuclear-receptor-mediated telomere insertion leads to genome instability in ALT cancers. *Cell* **160**, 913-927, doi:10.1016/j.cell.2015.01.044 (2015).
- 17 Attiyeh, E. F. *et al.* Chromosome 1p and 11q deletions and outcome in neuroblastoma. *The New England journal of medicine* **353**, 2243-2253, doi:10.1056/NEJMoa052399 (2005).
- 18 Guo, C. *et al.* Allelic deletion at 11q23 is common in MYCN single copy neuroblastomas. *Oncogene* **18**, 4948-4957, doi:10.1038/sj.onc.1202887 (1999).
- 19 Cubiles, M. D. *et al.* Epigenetic features of human telomeres. *Nucleic acids research* **46**, 2347-2355, doi:10.1093/nar/gky006 (2018).
- 20 Capurso, D., Xiong, H. & Segal, M. R. A histone arginine methylation localizes to nucleosomes in satellite II and III DNA sequences in the human genome. *BMC genomics* **13**, 630, doi:10.1186/1471-2164-13-630 (2012).

REVIEWER COMMENTS

Reviewer #2 (Remarks to the Author):

The authors have provided reasonable answers to my questions. I support the acceptance of this manuscript for publication.

Reviewer #3 (Remarks to the Author):

The paper entitled "Alternative lengthening of telomeres in childhood neuroblastoma from genome to proteome" by Hartlieb et al. has now been revised by the authors and I am satisfied with this new version because it has addressed all points raised during my precedent review. I appreciate the fact that all points raised have led to the generation of new data as well as new illustrations.

Reviewer #4 (Remarks to the Author):

The revised manuscript has satisfactorily addressed all my (minor) criticisms, and the authors also did their best to respond to other referee's comments. I am supporting the publication of this fine work in Nat. Com.